# DEEP VARIATIONAL INFERENCE SYMBOLIC REGRESSION

## ABSTRACT

A Bayesian inference approach to symbolic regression offers a combination of two powerful interpretability properties. On its own, symbolic regression offers explainable, unconstrained, closed-form expressions. However, combined with Bayesian inference, symbolic regression provides probability distributions over these interpretable models, accounting for real-world, limited, noisy data. Deep Symbolic Regression (DSR) is an algorithm that uses neural networks to perform symbolic regression; however, it aims to locate a *single* expression that best fits the data, rather than to calculate posteriors. In this work, we introduce Deep Variational Inference Symbolic Regression (DVISR). DVISR extends DSR into a fully Bayesian approach to symbolic regression by replacing the reward function used to train the network with the inner part of the expectation of the evidence lower bound. DVISR also modifies the architecture of the network to output probability distributions over constants within the expressions. This architectural modification enables the computation of the posterior distributions over both the expression trees and the constants contained within them. We show that DVISR can recover the true posterior distribution in simple settings and demonstrate scaling properties as the expression sizes get larger.

## 1 INTRODUCTION

Symbolic regression (SR) is the process of determining the mathematical relationships between independent and dependent variables using unconstrained closed-form expressions. When performing regression, typically, a dataset is provided in the form $(X, y)$, where $X \in \mathbb{R}^{n \times m}$, $y \in \mathbb{R}^m$, $n$ is the number of independent variables and $m$ is the number of data points. Regression aims to recover a function $f : \mathbb{R}^n \to \mathbb{R}$ that best explains the dataset. Many regression techniques restrict the functional form of $f$. However, it is usually difficult to know the functional form of the model apriori, thus, restricting the form can prevent the true model from being discovered. Symbolic regression removes constraints on the functional form of $f$, thereby considering expressions of multiple forms simultaneously.

One common feature of most symbolic regression algorithms is the ability to restrict the expression size. Thus, the output of these algorithms consists of *comprehensible* closed-form expressions restricted only by size, not by functional form. As a result, the expressions output by symbolic regression are expressive and typically interpretable, unlike black-box models such as neural networks. These properties make symbolic regression an attractive algorithm to use in safety-critical applications, such as healthcare or defence, where full transparency and explainability in decision-making are of paramount importance.

Multiple methods for performing symbolic regression have been successfully realised. Genetic programming (GP) techniques (Koza (1992)) create and optimise a population of expressions using an evolutionary algorithm (EA). This EA applies evolutionary operators, such as mutation, crossover and selection to trees that represent arbitrary functions. Other methods, such as Deep Symbolic Regression (DSR) (Petersen et al. (2021)), optimise neural networks that output arbitrary closed-form expressions.

Although GP and DSR successfully locate expressions that best explain the data, they do not aim to infer a probability distribution over expressions. Bayesian inference is a statistical reasoning technique that can construct and update a probability distribution over expressions, representing a degree

of belief over them. This formulation incorporates uncertainty about which model best explains the data, thereby accounting for noisy, limited, real-world data and providing more interpretable outputs.

The combination of symbolic regression and Bayesian inference has the potential to produce models that are significantly more interpretable. Markov-chain Monte Carlo (MCMC), a widespread method for performing Bayesian inference, has already been applied to symbolic regression (Jin et al. (2020); Guimerà et al. (2020); Xu et al. (2021); Zhao & Zhao (2025)). Although MCMC symbolic regression has the ability to accurately draw samples from the posterior, MCMC struggles to scale to higher dimensions and is generally slower than other methods for performing Bayesian inference, such as variational inference (Blei et al. (2017); Salimans et al. (2015); Gunapati et al. (2022).

Variational inference (VI) approximates the true posterior by minimising the Kullback-Leibler (KL) divergence between the true posterior and some proposed distribution via an optimisation procedure. VI is generally faster than MCMC and has the ability to scale to higher dimensions. However, the parametric form of the variational distribution limits how well it can match the true posterior, which can be complex in real-world applications. Nevertheless, VI and MCMC both have trade-offs rendering them appropriate for different use cases.

To the best of the authors' knowledge, there are only two algorithms that approach symbolic regression using variational inference. Fronk et al. (2024) perform variational inference over the weights of a polynomial neural network, from which polynomial expressions can be derived. However, this architecture limits the outputs to the class of polynomials. Sommerfelt & Hubin (2024) iteratively performs variational inference on subsets of the overall expression space, whilst also limiting the parameters over which Bayesian inference is carried out, due to computational intractability.

In this work, we add to this suite of algorithms that perform variational inference symbolic regression by introducing Deep Variational Inference Symbolic Regression (DVISR), an extension of DSR. Similar to DSR (Petersen et al. (2021)), DVISR uses a recurrent neural network to output a probability distribution over arbitrary tokens that is sampled from to create expressions. DVISR similarly uses REINFORCE, a policy gradient method, to train this network. However, rather than setting the reward to be the normalised root-mean-squared-error (NRMSE) bounded by a squashing function, we set the reward to be the inner part of the expectation of the evidence lower bound (ELBO). Therefore, as we maximise this altered reward via REINFORCE, we indirectly minimise the KL divergence between the probability distribution output by the neural network and the true posterior.

Furthermore, we also introduce a way to represent distributions over the constants within the expressions themselves, which are also optimised using variational inference. We do this by extending the outputs of the network to include parameters of distributions for each constant being sampled. This is in contrast to DSR where the constants are optimised according to the NRMSE for each expression. DVISR therefore performs variational inference over the *full* expression space and can include arbitrary operators in the expressions, resulting in a fully Bayesian approach to symbolic regression.

We demonstrate that DVISR correctly converges to the true posterior in a series of simple experiments. These experiments demonstrate the ability of DVISR to build posteriors over the constants in the expressions as well as over all expression trees. We also show how DVISR responds to an increasingly large expression size when constants are not included in the expressions.

Our contributions can be summarised as follows: (1) We introduce a novel algorithm for performing full variational inference symbolic regression, DVISR. (2) We demonstrate the ability of DVISR to converge to the true posterior on a number of simple experiments. (3) We explore the scalability of the algorithm and report the expression sizes at which DVISR becomes less effective.

## 2 METHODS

Our approach, Deep Variational Inference Symbolic Regression (DVISR) is an extension of the Deep Symbolic Regression algorithm (Petersen et al. (2021)). Therefore, much of the methods section illustrates the aspects of DSR that are included in DVISR. We explicitly state where DVISR departs from DSR.

## 2.1 EXPRESSION GENERATION

DVISR samples closed-form symbolic expressions using a recurrent neural network (RNN). This process is illustrated in Figure 1. The RNN outputs a categorical distribution over a library of tokens (Figure 1B) that corresponds to a customisable set of operators and variables. The RNN represents this probability distribution by outputting logits, which are subsequently input into a softmax function. The operators can take the form of binary operators, such as $\times$ and $+$; unary operators, such as trigonometric functions, the logarithm, and the exponential; independent variables, such as $x_0$ and $x_1$; and real valued constants.

Unlike DSR, DVISR additionally outputs a probability distribution over constant values. If a constant token is sampled in DSR, its value is determined downstream by a separate optimisation procedure. However, in DVISR this value is sampled from a distribution whose parameters are also output by the RNN. In this work, the RNN outputs the mean and variance of a normal distribution, but other types of distributions can be used.

The sampling procedure outputs a finite set of tokens and constant values that correspond to the postfix notation of a resulting symbolic expression. As such, one can also conceptualise the expression as a tree (Figure 1C). An expression tree is a representation of an expression where the nodes are operators and the leaves (terminating nodes) are variables or constants.

In order to assist the RNN with sampling, both the parent and the sibling of the current token being sampled are provided as input to the network, if they exist. However, there are situations in which only providing the parent and sibling as input to the network does not allow the network to see *all* previous tokens. Therefore, unlike DSR, DVISR also provides the previous token as input to the network. Tokens are provided as input in the form of a one-hot encoding. DVISR also provides the values of the respectively sampled constants as input.

Calculation of the likelihood, $q_\phi(f)$, of each expression, $f$, under the RNN is required in order to calculate the reward of the expression (as will be seen in Equation 5 of Section 2.2). The likelihood of each token, $\tau_i$, at position $i$ in the expression is given by the categorical distribution over tokens output by the RNN conditioned on all previous tokens and constants, $q_\phi(\tau_i|\tau_{1:i-1}, c_{1:i-1})$, where $\tau_{1:i-1}$ and $c_{1:i-1}$ are all previously sampled tokens and constants respectively. The likelihood of each constant, $c_i$, at position $i$ in the expression is given by the normal distribution over constants output by the RNN conditioned on all previous tokens and constants, $q_\phi(c_i|\tau_{1:i-1}, c_{1:i-1})$. Therefore, the likelihood of the expression, $f$, is the product of the likelihood of the tokens and constants that are included in the expression:

$$q_\phi(f) = \prod_{i=1}^{L} \begin{cases} q_\phi(\tau_i|\tau_{1:i-1}, c_{1:i-1}) \cdot q_\phi(c_i|\tau_{1:i-1}, c_{1:i-1}) & \text{if } \tau_i = \text{const} \\ q_\phi(\tau_i|\tau_{1:i-1}, c_{1:i-1}) & \text{otherwise} \end{cases} \tag{1}$$

where $L$ is the number of tokens in the expression.

Like in DSR, DVISR also imposes constraints on certain properties of the expressions whilst sampling. Examples of such constraints include: a maximum expression size; no child whose inverse is its parent, e.g. $\exp(\log(x))$; and no nested trigonometric functions, e.g. $\sin(x + \cos(x))$. These constraints are imposed at the network level by eliciting a zero sampling probability for tokens that would cause the constraints to be violated.

## 2.2 TRAINING PROCEDURE

As in DSR, the RNN in DVISR is trained using the policy gradient method REINFORCE (Williams (1992)). Both DSR and DVISR embed symbolic regression in this framework by conceptualising: (i) sampled tokens as actions, (ii) parent, sibling and previous tokens as states, (iii) the RNN output, $q_\phi$, as a policy, and (iv) a sequence of tokens as an episode. As such, in DSR and DVISR the aim is to maximise the expected reward of expressions drawn from the RNN policy:

$$J(\phi) = \mathbb{E}_{f \sim q_\phi(f)} [R(f)] \tag{2}$$

where $f$ is an expression sampled from the policy, $q_\phi(f)$, parameterised by $\phi$ and $R(f)$ is some reward function computed on that expression.

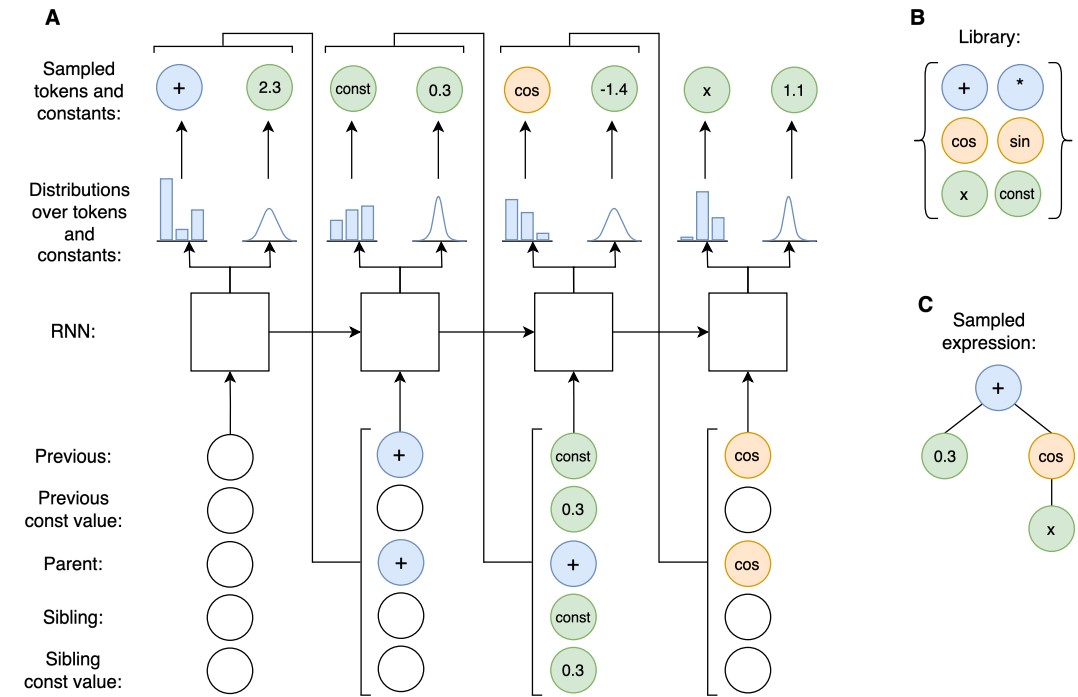

Figure 1: **A**: An example of the sampling procedure of DVISR. The previous, parent, sibling and respective constant value tokens of the about to be sampled token are provided as input to the RNN. The RNN outputs parameters of both a categorical distribution over tokens and a normal distribution over constants. The respective token and constant are sampled and are included in the expression (**C**). The constant value is only included in the expression if a *const* token was sampled. This process repeats until the maximum number of tokens has been sampled or until the expression tree is complete. **B**: The extensible token library from which all elements can be sampled. **C**: The resultant sampled expression tree which corresponds to the following closed-form expression: $0.3 + \cos(x)$. This figure was inspired by the illustration in Petersen et al. (2021) for ease of comparison.

Maximisation of Equation 2 is performed via gradient ascent using the REINFORCE policy gradient:

$$\nabla_\phi J(\phi) = \mathbb{E}_{f \sim q_\phi(f)}[\nabla_\phi \log q_\phi(f) \cdot R(f)] \tag{3}$$

In most instances, the gradients $\nabla_\phi J(\phi)$, cannot be calculated exactly because the expectation is taken over an insurmountably large discrete function space, or an infinite continuous function space. Therefore, an estimation of the expectation is calculated by computing the sample mean over a batch of $N$ expressions:

$$\nabla_\phi J(\phi) \approx \frac{1}{N} \sum_{i=1}^{N} [\nabla_\phi \log q_\phi(f_i) \cdot R(f_i)] \tag{4}$$

This gradient estimate is unbiased but often has high variance. To reduce this variance, it is common practice to subtract a baseline value from the reward function. In this work, both the mean and the exponentially weighted moving average of the rewards of the batch are used as baselines, depending on the experiment.

In order to locate the singular model that best fits the data, DSR sets the reward function to a squashed version of the normalised root-mean squared error between the data and the model prediction. However, in DVISR, the reward function is set to the inside term of the evidence lower bound (ELBO) expectation:

$$R(f) = \log \mathcal{L}_\theta(f|X, y) + \log p(f) - \log q_\phi(f) \tag{5}$$

where $f$ is the considered expression (or model); $\mathcal{L}_\theta(f|X,y)$ is the likelihood of the model, $f$, given the dataset, $(X,y)$; and $\theta$ are the parameters of the likelihood function. In this work, we let $\mathcal{L}_\theta(f|X,y) = \prod_{i=1}^{n} \mathcal{N}(y_i; f(X_i), \sigma_L^2)$, where $y_i$ and $X_i$ refer to the values of the $i$-th data point, $n$ is the number of data points in the dataset, and $\sigma_L$ is the standard deviation of the likelihood function. $p(f)$ is the prior for a particular expression. $q_\phi(f)$ is the likelihood of the expression under the variational distribution, as described in Equation 1.

By substituting the reward function (Equation 5) into Equations 2 and 3 we derive the the final objective to maximise and its policy gradient respectively:

$$J(\phi) = \text{ELBO} = \mathbb{E}_{f \sim q_\phi(f)}[\log \mathcal{L}_\theta(f|X,y) + \log p(f) - \log q_\phi(f)] \tag{6}$$

$$\nabla_\phi J(\phi) = \mathbb{E}_{f \sim q_\phi(f)}[\nabla_\phi \log q_\phi(f) \cdot (\log \mathcal{L}_\theta(f|X,y) + \log p(f) - \log q_\phi(f))] \tag{7}$$

In DVISR the aim is to approximate the true posterior, $p(f|X,y)$, using the variational distribution $q_\phi(f)$. The process of variational inference achieves this by maximising the ELBO (Equation 6) with respect to $\phi$, utilising the fact that:

$$\log p(y|X) = \text{ELBO} + \text{KL}(q_\phi(f)||p(f|X,y)) \tag{8}$$

where $p(y|X)$ is the evidence and $\text{KL}(q_\phi(f)||p(f|X,y))$ is the KL divergence between the variational distribution and the true posterior. The evidence is a constant; therefore, Equation 8 makes it clear that maximising the ELBO will minimise the KL divergence. It is now self evident that by maximising Equation 6 the KL divergence between the variational distribution and the true posterior will be minimised. A proof of the equality in Equation 8 is given in Appendix A.1.

Unlike DSR, DVISR does not use the risk-seeking policy gradient, as doing so would result in the maximisation of a different objective than Equation 6, thereby invalidating the variational inference approach.

## 3 EXPERIMENTS

We present two main categories of experiments: those with and those without a constant token in the token library. When the constant is not included in the token library, the RNN only outputs a categorical distribution over tokens.

### 3.1 WITHOUT CONSTANTS

#### 3.1.1 SIMPLE EXPERIMENTS

We first present three simple experiments that illustrate all the desired properties of DVISR. In these experiments, we generate data (without noise) from $y = x_0^2$; $y = x_0$; and $y = 0.5$ respectively. The values of $x_0$ are selected from the range $[0.0, 1.0]$ at intervals of $0.1$, resulting in 11 data points in total. The token library is set to be $\{+, *, \sin, x_0\}$, which includes both unary and binary operators. The maximum number of tokens in an expression is set to 3, and a constraint on nested trigonometric functions is applied, as in Petersen et al. (2021). These constraints mean that there are only 4 possible expressions that adhere to these constraints: $y = x_0$, $y = x_0 * x_0$, $y = x_0 + x_0$ and $y = \sin(x_0)$. Therefore, the true expression is locatable in the search space for two of the experiments, $y = x_0^2$ and $y = x_0$, but not in the case of $y = 0.5$. The prior, $p(f)$, is set to be the uniform prior over all possible expression trees that can be generated according to the token library and the maximum expression size constraint (that is, $\frac{1}{5}$ for each expression in this case - the four previously mentioned and $y = \sin(\sin(x_0))$, which is not allowed due to the constraint on nested trigonometric functions). The optimisation procedure sampled 100 equations every epoch for 250 epochs. This procedure was ran 10 times. A full list of hyperparameters is provided in Table 7 in Appendix A.6.

Figure 2 illustrates the median and the inter-quartile range (IQR) of the ELBO (blue) over 10 runs for the simple no constant experiment where data is generated from $y = x_0^2$. Figure 2 shows how the ELBO converges to the log of the evidence (orange). Large initial gains are shown within the first 25 epochs, with the remaining epochs dedicated to improved precision (magnified). A learning rate annealer, described in Table 7, is used to elicit this high precision.

In these experiments, the evidence was calculated by application of the law of total probability over the 4 aforementioned possible expressions:

$$p(y|X) = \sum_{i=1}^{4} p(y, f_i|X) \quad (9)$$

which by application of the chain rule, assuming $p(f_i|X) = p(f_i)$, and rewriting $p(y|X, f_i)$ as $\mathcal{L}_\theta(f_i|X, y)$ can be expressed as:

$$p(y|X) = \sum_{i=1}^{4} \mathcal{L}_\theta(f_i|X, y) \cdot p(f_i)$$
$$(10)$$

The calculation of the evidence is usually intractable which justifies the need for variational inference. However, in this instance, the model space is finite and small, which means that the evidence - and by extension, the true posterior values - can be calculated directly and used to validate the effectiveness of DVISR.

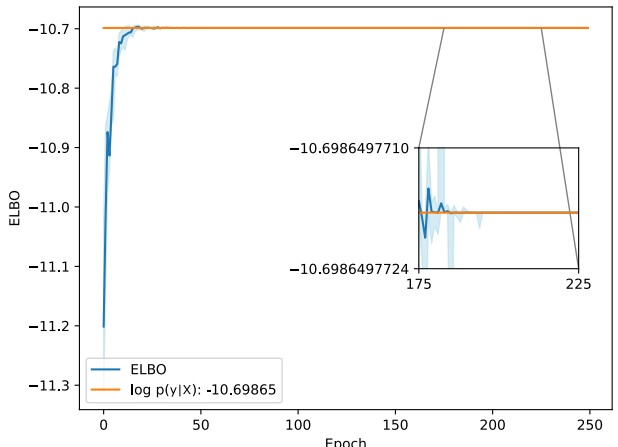

Figure 2: The median and IQR of the ELBO (Eq. 6) values over 10 runs for the simple no constant token experiment where data is generated from $y = x_0^2$ (blue). The log of the evidence is shown in orange.

Figure 6 in Appendix A.4.1 illustrates how the KL divergence, calculated via a simple rearrangement of Equation 8, approaches zero as the optimisation procedure progresses. This convergence happens in conjunction with the ELBO approaching the log of the evidence. This behaviour is predicted by the relations derived in Equation 8 and confirms that the variational distribution, $q_\phi(f)$, has successfully converged to the true posterior, $p(f|X, y)$. As further evidence of correct convergence, Table 1 reports the true posteriors and variational distribution median and IQR values for all possible expressions over 10 runs for the experiment where data is generated from $y = x_0^2$. The respective figures and results tables are also provided in the Appendix for the other two experiments where data is generated from $y = x_0$ (Figures 4 and 7 and Table 3) and $y = 0.5$ (Figures 5 and 8 and Table 4). Very similar behaviour to the experiment where data is generated from $y = x_0^2$ is shown.

Table 1: True posterior, $p(f|X, y)$, and median and IQR of variational probability, $q_\phi(f)$, for all possible expressions over 10 runs for the simple no constant token experiment where data is generated from $y = x_0^2$. Values are rounded to eight decimal places.

| Expression | $p(f|X, y)$ | $q_\phi(f)$ (median + IQR) | |
|---|---|---|---|
| $x_0 * x_0$ | 0.36091529 | 0.36091529 | [0.36091529, 0.36091529] |
| $\sin(x_0)$ | 0.31404061 | 0.31404061 | [0.31404061, 0.31404061] |
| $x_0$ | 0.30551329 | 0.30551329 | [0.30551329, 0.30551329] |
| $x_0 + x_0$ | 0.01953081 | 0.01953081 | [0.01953081, 0.01953081] |

### 3.1.2 SCALING EXPERIMENTS

We now present a series of experiments that demonstrates the scalability of DVISR. As before the token library is set to $\{+, *, \sin, x_0\}$ and the same constraints are used. However, we vary the maximum number of tokens that can be present in an expression from 1 to 12. The aim is to elicit how DVISR performs as the search space increases in size. Here, we use data generated only from $y = x_0^2$, where the values of $x_0$ are selected from the range $[0.0, 1.0]$ at intervals of 0.1, as before. The problem difficulty in this setting is mostly determined by the maximum expression size rather than the underlying data. We therefore chose to keep the data constant and to vary the maximum

expression size. The optimisation procedure sampled 1000 equations every epoch for 2000 epochs, which was repeated for maximum expression sizes ranging from 1 to 12. The hyperparameters were altered slightly to account for larger expression sizes (Table 8 in Appendix A.6).

Figure 3 illustrates how the KL divergence between the true posterior and the *optimised* variational distribution increases as the maximum expression size increases. 50 000 expressions were sampled from the final variational distribution and used to estimate the ELBO, from which the KL divergence was calculated using a rearrangement of Equation 8. Experiments involving maximum expression sizes greater than 12 were not performed due to the computational difficult of calculating the evidence at that scale.

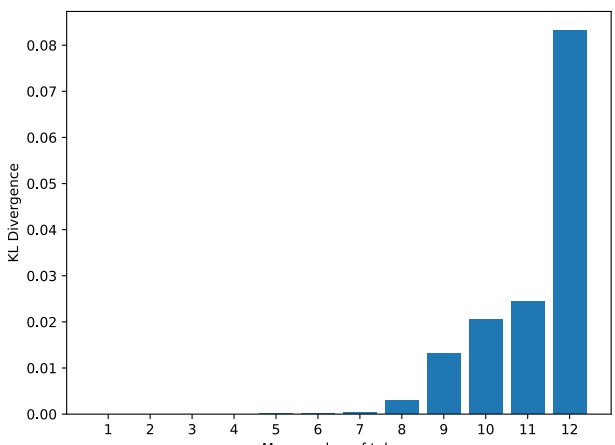

Figure 3: Estimated KL divergence between the true posterior and the optimised variational distribution according to 50000 samples for a range of maximum expression sizes.

## 3.2 WITH CONSTANTS

We present three simple experiments that demonstrates the abilities of DVISR when constants *are* included in the token library. As with the simple no constant token experiments non-noisy data is generated from $y = x_0^2$, $y = x_0$ and $y = 0.5$, resulting in three separate experiments. The values of $x_0$ are selected from the range $[0.0, 1.0]$ at intervals of 0.1 resulting in 11 data points in total. The token library is set to be $\{+, *, \cos, c, x_0\}$, where $c$ is the constant token. The maximum number of tokens in an expression is set to 3. As in Petersen et al. (2021), constraints are applied to prevent nested trigonometric functions and to prevent all children of an operator from being constants, since this would simply result in another constant. An additional constraint forcing the left hand side child of binary operators to be a constant is also applied. This constraint reduces the number of equivalent expressions, such as $y = c * x$ and $y = x * c$. Due to these constraints there are 7 allowed distinct expression trees (highlighted in Table 2); however, due to the fact that constant tokens are also considered, there is technically an infinite number of total expressions considered in this experiment. The optimisation procedure sampled 500 equations every epoch for 1000 epochs. This procedure was ran 10 times. A full list of hyperparameters is provided in Table 9 in Appendix A.6.

The prior, $p(f)$, is defined as:

$$p(f) = \frac{1}{N_{expr}} \prod_{i=1}^{N_c} \mathcal{N}(c_i; \mu_c, \sigma_c^2) \tag{11}$$

where $N_{expr}$ is the number of expression trees that can be constructed using the token library with a maximum number of tokens of 3, $N_c$ is the number of constants contained in $f$, $c_i$ is the $i$-th constant in $f$, $\mu_c$ is the prior mean over $c$ and $\sigma_c$ is the prior standard deviation over $c$.

Table 2 illustrates how the variational distribution, $q_\phi(z)$, converges to the exact true posterior, $p(z|X, y)$, to eight decimal places for this simple experiment with constant tokens. Here, $z$ refers to the particular expression tree within $f$. When constants are included, $f$ can be considered as being composed of an expression tree, $z$, and constant values within that particular tree, $\mathbf{c}$. The probabilities $q_\phi(f)$ and $p(f|X, y)$ actually refer to $q_\phi(z, \mathbf{c})$ and $p(z, \mathbf{c}|X, y)$ respectively, where $\mathbf{c}$ are the constant values contained within the particular $f$. The values reported in Table 2 where calculated by marginalising out the constant values from $p(z, \mathbf{c}|X, y)$ and $q_\phi(z, \mathbf{c})$ with a numerical integrator. Tables 5 and 6 in the Appendix show equivalent results for the experiments where data is generated from $y = x_0^2$ and $y = 0.5$ respectively.

Table 2: True posterior, $p(z|X, y)$ and median and IQR of variational probability, $q_\phi(z)$, for all possible expressions over 10 runs for the simple constant experiment where data is generated from $y = x_0^2$. Values are rounded to eight decimal places.

| Expression | $p(z|X, y)$ | $q_\phi(z)$ (median + IQR) |
|---|---|---|
| $x_0 * x_0$ | 0.48299064 | 0.48299064 | [0.48299064, 0.48299064] |
| $x_0$ | 0.40884956 | 0.40884956 | [0.40884956, 0.40884956] |
| $\cos(x_0)$ | 0.03737588 | 0.03737588 | [0.03737588, 0.03737588] |
| $x_0 + x_0$ | 0.02613688 | 0.02613688 | [0.02613688, 0.02613688] |
| $x_0 * c$ | 0.02266321 | 0.02266321 | [0.02266321, 0.02266321] |
| $x_0 + c$ | 0.01394328 | 0.01394328 | [0.01394328, 0.01394328] |
| $c$ | 0.00804056 | 0.00804056 | [0.00804056, 0.00804056] |

In order to calculate the true posterior, $p(z, \mathbf{c}|X, y)$, the evidence, $p(y|X)$, was first calculated using a numerical integrator. The evidence was calculated in the following way:

$$p(y|X) = \sum_z \int_{-\infty}^{\infty} \ldots \int_{-\infty}^{\infty} \mathcal{L}_\theta(z, c_1, \ldots, c_N|X, y) \cdot p(z, c_1, \ldots, c_N) \, dc_1 \ldots dc_N \quad (12)$$

where $f$ has been split into its component parts: $z$, which is the particular expression tree, and $c_1 \ldots c_N$, are the constant values in *all* of the possible expression trees. Equation 12 calculates the evidence by marginalising out all of the possible expression trees, $z$, and constants $c_1, \ldots, c_N$.

Naively solving this integration using a numerical solver does not result in the correct value of the evidence. Firstly, we must define what it means to calculate $\int_{-\infty}^{\infty} \mathcal{L}_\theta(z, c_1, \ldots, c_N|X, y) \cdot p(z, c_1, \ldots, c_N) \, dc_i$, where $c_i \notin z$. $c_i \notin z$ denotes the situation where the particular constant $c_i$ is not contained in the particular expression tree, $z$; this situation regularly occurs as part of Equation 12. The value $\mathcal{L}_\theta(z, c_1, \ldots, c_N|X, y) \cdot p(z, c_1, \ldots, c_N)$, where $c_i \notin z$, is constant with respect to $c_i$. However, any definite integral of a non-zero constant over an infinite domain diverges, and would result in Equation 12 evaluating to $p(y|X) = \infty$, which is clearly incorrect. Instead we choose to define $\int_{-\infty}^{\infty} \mathcal{L}_\theta(z, c_1, \ldots, c_N|X, y) \cdot p(z, c_1, \ldots, c_N) \, dc_i$, where $c_i \notin z$, as simply $\mathcal{L}_\theta(z, c_1, \ldots, c_N|X, y) \cdot p(z, c_1, \ldots, c_N)$. In Appendix A.2, we prove that this definition results in the correct evidence being calculated.

Progression of the ELBO throughout the optimisation process is illustrated in the Appendix in Figures 9, 10 and 11 for the experiments where data is generated from $y = x_0^2$, $y = x_0$ and $y = 0.5$ respectively. The behaviour is very similar to that of the no constant token experiments where large initial gains are made in the early epochs and the remaining epochs are dedicated to fine-tuning the precision of the variational distribution. As with the no constant token experiments, the evolution of the KL divergence over the optimisation process is also shown in the Appendix in Figures 12, 13 and 14.

## 4 RELATED WORK

Deep learning based-symbolic regression is an area of research that has seen significant developments in recent years. RNN-based approaches include Petersen et al. (2021), who introduced Deep Symbolic Regression (DSR) and Landajuela et al. (2022) who proposed a unified framework that combines RNN-based generation with other SR strategies. Bastiani et al. (2024) extends DSR (Petersen et al. (2021)) by replacing its reward function with the Bayesian Information Criterion, which penalises complexity, and Popov et al. (2023) uses a variational autoencoder to generate symbolic expressions according to user-defined constraints. Transformer-based models include: an end-to-end approach that generates full symbolic expressions, including constants, in a single forward pass (Kamienny et al. (2022)); planning-based methods, which combine a pre-trained transformer with Monte Carlo Tree Search to better balance expression accuracy and complexity (Shojaee et al. (2023)); and a transformer pre-trained to capture both equation and dataset invariances, before fine-tuning on target tasks (Holt et al. (2023)). Grayeli et al. (2024) introduced a library of reusable symbolic *concepts* via LLM-assisted evolutionary operations, such as mutations and crossovers. Fully-connected architectures have also been considered (Kim et al. (2020); Zhang et al. (2023)),

which replace conventional activations with symbolic functions, thereby searching over a space of expressions. Hybrid neural-symbolic approaches have also been explored: AI Feynman (Udrescu & Tegmark (2020); Udrescu et al. (2020)) uses a fully connected neural network to detect symmetries and separable structures, enabling recursive symbolic simplification; and Mundhenk et al. (2021) pair an RNN with GP, using the former to seed the population.

Bayesian approaches to SR offer an alternative by explicitly modelling uncertainty over multiple symbolic expressions. Both Guimerà et al. (2020) and Jin et al. (2019) employ MCMC to search the space of expression trees and estimate the posterior. Xu et al. (2021) proposes a Bayesian expectation-maximisation framework that alternates between inferring latent physical properties via MCMC and discovering symbolic force laws using grammar-constrained regression. Zhao & Zhao (2025) adopts a two stage approach: GP is first used to generate a symbolic expression, then MCMC is applied to infer a posterior over the expression's parameters. Ellis et al. (2018) proposes $EC^2$, which iteratively refines a frontier of high-likelihood program primitives for a *set* of tasks, and a neural network induces a probability distribution over this frontier given a particular task, which guides search. Despite the neural network being trained in a way similar to variational inference, it does not output a probability over all programs, only those in the frontier. Furthermore, unlike DVISR, $EC^2$ has no method of defining a probability distribution over constants within expressions, rather, it subsequently optimises them using gradient descent.

Variational inference has previously been applied to symbolic regression via a polynomial neural network architecture (Fronk et al. (2024)). The parameters of this network correspond to the coefficients of a resulting polynomial expression. Unlike DVISR, the work of Fronk et al. (2024) is restricted to fixed-degree polynomials. In Sommerfelt & Hubin (2024), evolutionary variational Bayes (EVB) is proposed, which iteratively performs variational inference on subsets of the space of all models. The models considered in EVB include arbitrary non-linear transformations, called features; therefore, EVB can be used for symbolic regression. EVB does not use a neural network to define probabilities over tokens, like DVISR; rather, it builds distributions over the coefficients of these features. Unlike with DVISR, the experiments in Sommerfelt & Hubin (2024) do not perform Bayesian inference over *all* the model parameters (leaving out the parameters $\alpha$) due to the issues with scalability (Hubin et al. (2022)).

## 5 CONCLUSION & FUTURE WORK

We introduce a novel algorithm, DVISR, by extending DSR into a fully Bayesian symbolic regression solution. Not only does DVISR calculate the posterior over expression trees, but it also calculates the posterior over all the constants within them. We show that in multiple simple experiments DVISR converges repeatably and precisely to the true posterior, both for cases with and without constant tokens in the token library. We also demonstrate the scalability of DVISR when constant tokens are not included in the token library.

The scaling experiments in Section 3.1.2 demonstrate that one of the main limitations of DVISR is a decreased performance as the expression size increases. This is unfortunate, but expected, as similar issues are reported in Hubin et al. (2022) when attempting to scale their full Bayesian solution. Future work could ameliorate these limitations by conducting a more extensive hyperparameter sweep; employing a more state-of-the-art architecture, such as a transformer (Kamienny et al. (2022); Shojaee et al. (2023); Holt et al. (2023)); or adopting a more advanced reinforcement learning algorithm, such as Proximal Policy Optimisation (Schulman et al. (2017)) or Group Relative Policy Optimisation (Shao et al. (2024)).

The main limitation of this study is the simplicity of the experiments, particularly the small expression sizes, the low number of independent variables, and the absence of real-world datasets. This leaves the question open of whether DVISR, or variants of it, will be sufficiently scalable to address real-world problems. However, we believe that DVISR demonstrates initial promise in producing highly explainable machine learning models through a novel approach.

# 6 REPRODUCIBILITY STATEMENT

In order to demonstrate the reproducibility of the results in this work, we have provided the full source code and results used for our experiments in the supplementary material. The source code provides instructions on how to install and set up an environment to run the exact experiments that were ran in this work. The data and experimental setup are provided in the paper (Section 3 and Appendix A.6) and in the configuration files in the source code.

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

# A APPENDIX

## A.1 ELBO, KL DIVERGENCE AND EVIDENCE EQUALITY PROOF

Here, we prove Equation 8, which is (as a reminder):

$$\log p(y|X) = \text{ELBO} + \text{KL}(q_\phi(f)||p(f|X, y)) \tag{8}$$

and in expanded form is:

$$\log p(y|X) = \mathbb{E}_{f \sim q_\phi(f)}[\log \mathcal{L}_\theta(f|X, y) + \log p(f) - \log q_\phi(f)] + \text{KL}(q_\phi(f)||p(f|X, y)) \tag{13}$$

We begin by expanding the definition of the KL divergence:

$$\text{KL}(q_\phi(f)||p(f|X, y)) = \mathbb{E}_{f \sim q_\phi(f)} \left[ \log \frac{q_\phi(f)}{p(f|X, y)} \right] \tag{14}$$

and applying the quotient rule for logarithms:

$$\text{KL}(q_\phi(f)||p(f|X, y)) = \mathbb{E}_{f \sim q_\phi(f)} \left[ \log q_\phi(f) - \log p(f|X, y) \right] \tag{15}$$

We now expand the posterior using Bayes rule for three events:

$$\text{KL}(q_\phi(f)||p(f|X, y)) = \mathbb{E}_{f \sim q_\phi(f)} \left[ \log q_\phi(f) - \log \frac{p(y|X, f)p(f|X)}{p(y|X)} \right] \tag{16}$$

and then apply the product and quotient rule for logarithms:

$$\text{KL}(q_\phi(f)||p(f|X, y)) = \mathbb{E}_{f \sim q_\phi(f)} \left[ \log q_\phi(f) - (\log p(y|X, f) + \log p(f|X) - \log p(y|X)) \right] \tag{17}$$

We can simplify by distributing the minus throughout the brackets:

$$\text{KL}(q_\phi(f)||p(f|X, y)) = \mathbb{E}_{f \sim q_\phi(f)} \left[ \log q_\phi(f) - \log p(y|X, f) - \log p(f|X) + \log p(y|X) \right] \tag{18}$$

We can bring the log of the evidence, $\log p(y|X)$, outside of the expectation because it is constant with respect to $f$:

$$\text{KL}(q_\phi(f)||p(f|X, y)) = \mathbb{E}_{f \sim q_\phi(f)} \left[ \log q_\phi(f) - \log p(y|X, f) - \log p(f|X) \right] + \log p(y|X) \tag{19}$$

At this point it is worth noting that $\log p(y|X, f)$ is the likelihood and can therefore be written as such. Also, $\log p(f|X)$ is the prior over $f$, which is independent of $X$. Therefore, from now on, we will write $p(f|X)$ as simply $p(f)$:

$$\text{KL}(q_\phi(f)||p(f|X, y)) = \mathbb{E}_{f \sim q_\phi(f)} \left[ \log q_\phi(f) - \log \mathcal{L}_\theta(f|X, y) - \log p(f) \right] + \log p(y|X) \tag{20}$$

Finally, we factorise the minus out of the expectation and rearrange to arrive at Equation 13:

$$\log p(y|X) = \mathbb{E}_{f \sim q_\phi(f)} \left[ \log \mathcal{L}_\theta(f|X, y) + \log p(f) - \log q_\phi(f) \right] + \text{KL}(q_\phi(f)||p(f|X, y))$$

which is equivalent to Equation 8:

$$\log p(y|X) = \text{ELBO} + \text{KL}(q_\phi(f)||p(f|X, y))$$

$$\square$$

## A.2 NUMERICAL INTEGRATOR EVIDENCE PROOF

In order to calculate the evidence, $p(y|X)$, all aspects of the model, $f$, must be marginalised out. $f$ can be split into separate components: $z$, the particular expression tree, and $c_1 \ldots c_N$, the continuous constants within *all* of the expression trees. The evidence can then be calculated in the following way:

$$p(y|X) = \sum_z \int_{-\infty}^{\infty} \ldots \int_{-\infty}^{\infty} \mathcal{L}_\theta(z, c_1, \ldots, c_N | X, y) \cdot p(z, c_1, \ldots, c_N) \, dc_1 \ldots dc_N \tag{21}$$

We are now interested in how to define each integral on the right hand side of Equation 21, such that it is equivalent to $p(y|X)$.

There are two distinct types of integrals here: one when the constant being integrated is in the expression tree, $z$, and the other, when the constant being integrated is *not* in the expression tree, $z$. For clarity let's separate all the constants into two sets for each $z$: $\mathbf{c}_{\in z}$ for those within $z$ and $\mathbf{c}_{\notin z}$ for those not in $z$. We can then rewrite the right hand side of Equation 21 as follows:

$$\sum_z \int_{-\infty}^{\infty} \int_{-\infty}^{\infty} \mathcal{L}_\theta(z, \mathbf{c}_{\in z}, \mathbf{c}_{\notin z}|X, y) \cdot p(z, \mathbf{c}_{\in z}, \mathbf{c}_{\notin z}) \, d\mathbf{c}_{\notin z}, d\mathbf{c}_{\in z} \tag{22}$$

Let $\int_{-\infty}^{\infty} \mathcal{L}_\theta(z, \mathbf{c}_{\in z}, \mathbf{c}_{\notin z}|X, y) \cdot p(z, \mathbf{c}_{\in z}, \mathbf{c}_{\notin z}) \, d\mathbf{c}_{\notin z} = \mathcal{L}_\theta(z, \mathbf{c}_{\in z}|X, y) \cdot p(z, \mathbf{c}_{\in z})$. We can now simplify Equation 22 as follows:

$$\sum_z \int_{-\infty}^{\infty} \mathcal{L}_\theta(z, \mathbf{c}_{\in z}|X, y) \cdot p(z, \mathbf{c}_{\in z}) \, d\mathbf{c}_{\in z} \tag{23}$$

In most cases $\int_{-\infty}^{\infty} \mathcal{L}_\theta(z, \mathbf{c}_{\in z}|X, y) \cdot p(z, \mathbf{c}_{\in z}) \, d\mathbf{c}_{\in z}$ is not analytically solvable, therefore, in this work, we use a numerical solver to compute this integral. Solving this integral in the usual way marginalises out $\mathbf{c}_{\in z}$, resulting in:

$$\sum_z \mathcal{L}_\theta(z|X, y) \cdot p(z) \tag{24}$$

Rewriting the likelihood as a probability distribution over $y$ results in:

$$\sum_z p(y|X, z) \cdot p(z) \tag{25}$$

By applying the chain rule, and assuming that $p(z|X) = p(z)$, which is true in this work, we can simplify to:

$$\sum_z p(y, z|X) \tag{26}$$

Applying the law of total probability to Equation 26 results in $p(y|X)$, the correct value of the evidence.

Equating $\int_{-\infty}^{\infty} \mathcal{L}_\theta(z, \mathbf{c}_{\in z}, \mathbf{c}_{\notin z}|X, y) \cdot p(z, \mathbf{c}_{\in z}, \mathbf{c}_{\notin z}) \, d\mathbf{c}_{\notin z}$ to $\mathcal{L}_\theta(z, \mathbf{c}_{\in z}|X, y) \cdot p(z, \mathbf{c}_{\in z})$ is not an intuitive step. Given that $\mathcal{L}_\theta(z, \mathbf{c}_{\in z}, \mathbf{c}_{\notin z}|X, y) \cdot p(z, \mathbf{c}_{\in z}, \mathbf{c}_{\notin z})$ is a constant with respect to $\mathbf{c}_{\notin z}$ it would be natural to treat this integral in the usual way, resulting in a divergent integral. However, in that case, the right hand side of Equation 21 is not equivalent to $p(y|X)$. Only by letting $\int_{-\infty}^{\infty} \mathcal{L}_\theta(z, \mathbf{c}_{\in z}, \mathbf{c}_{\notin z}|X, y) \cdot p(z, \mathbf{c}_{\in z}, \mathbf{c}_{\notin z}) \, d\mathbf{c}_{\notin z} = \mathcal{L}_\theta(z, \mathbf{c}_{\in z}|X, y) \cdot p(z, \mathbf{c}_{\in z})$ do we compute the correct evidence. $\square$

### A.3 EXPERIMENT RESULTS

#### A.3.1 NO CONSTANT TOKEN EXPERIMENTS

Table 3: True posterior, $p(f|X, y)$ and median and IQR of variational probability, $q_\phi(f)$, for all possible expressions over 10 runs for the simple no constant experiment where data is generated from $y = x_0$. Values are rounded to eight decimal places.

| Expression | $p(f|X, y)$ | $q_\phi(f)$ (median + IQR) | |
|---|---|---|---|
| $x_0$ | 0.33699068 | 0.33699068 | [0.33699068, 0.33699068] |
| $\sin(x_0)$ | 0.32858934 | 0.32858934 | [0.32858934, 0.32858934] |
| $x_0 * x_0$ | 0.28526121 | 0.28526121 | [0.28526121, 0.28526121] |
| $x_0 + x_0$ | 0.04915877 | 0.04915877 | [0.04915877, 0.04915877] |

Table 4: True posterior, $p(f|X, y)$ and median and IQR of variational probability, $q_\phi(f)$, for all possible expressions over 10 runs for the simple no constant experiment where data is generated from $y = 0.5$. Values are rounded to eight decimal places.

| Expression | $p(f|X, y)$ | $q_\phi(f)$ (median + IQR) | |
|---|---|---|---|
| $\sin(x_0)$ | 0.37718952 | 0.37718952 | [0.37718952, 0.37718952] |
| $x_0$ | 0.32865058 | 0.32865058 | [0.32865058, 0.32865058] |
| $x_0 * x_0$ | 0.27820135 | 0.27820135 | [0.27820135, 0.27820135] |
| $x_0 + x_0$ | 0.01595856 | 0.01595856 | [0.01595856, 0.01595856] |

A.3.2    CONSTANT TOKEN EXPERIMENTS

Table 5: True posterior, $p(z|X, y)$ and median and IQR of variational probability, $q_\phi(z)$, for all possible expressions over 10 runs for the simple constant experiment where data is generated from $y = x_0$. Values are rounded to eight decimal places.

| Expression | $p(z|X, y)$ | $q_\phi(z)$ (median + IQR) | |
|---|---|---|---|
| $x_0$ | 0.44342029 | 0.44342029 | [0.44342029, 0.44342029] |
| $x_0 * x_0$ | 0.37535343 | 0.37535343 | [0.37535343, 0.37535343] |
| $\cos(x_0)$ | 0.07302076 | 0.07302076 | [0.07302076, 0.07302076] |
| $x_0 + x_0$ | 0.06468427 | 0.06468427 | [0.06468427, 0.06468427] |
| $x_0 * c$ | 0.02245722 | 0.02245722 | [0.02245722, 0.02245722] |
| $x_0 + c$ | 0.01336355 | 0.01336355 | [0.01336355, 0.01336355] |
| $c$ | 0.00770048 | 0.00770048 | [0.00770048, 0.00770048] |

Table 6: True posterior, $p(z|X, y)$ and median and IQR of variational probability, $q_\phi(z)$, for all possible expressions over 10 runs for the simple constant experiment where data is generated from $y = 0.5$. Values are rounded to eight decimal places.

| Expression | $p(z|X, y)$ | $q_\phi(z)$ (median + IQR) | |
|---|---|---|---|
| $x_0$ | 0.34938537 | 0.34938537 | [0.34938537, 0.34938537] |
| $x_0 * x_0$ | 0.29575326 | 0.29575326 | [0.29575326, 0.29575326] |
| $\cos(x_0)$ | 0.28838233 | 0.28838233 | [0.28838233, 0.28838233] |
| $x_0 * c$ | 0.02075641 | 0.02075641 | [0.02075641, 0.02075641] |
| $c$ | 0.01822765 | 0.01822765 | [0.01822765, 0.01822765] |
| $x_0 + x_0$ | 0.01696539 | 0.01696539 | [0.01696539, 0.01696539] |
| $x_0 + c$ | 0.01052958 | 0.01052958 | [0.01052958, 0.01052958] |

## A.4 EXPERIMENT FIGURES

### A.4.1 NO CONSTANT TOKEN EXPERIMENTS

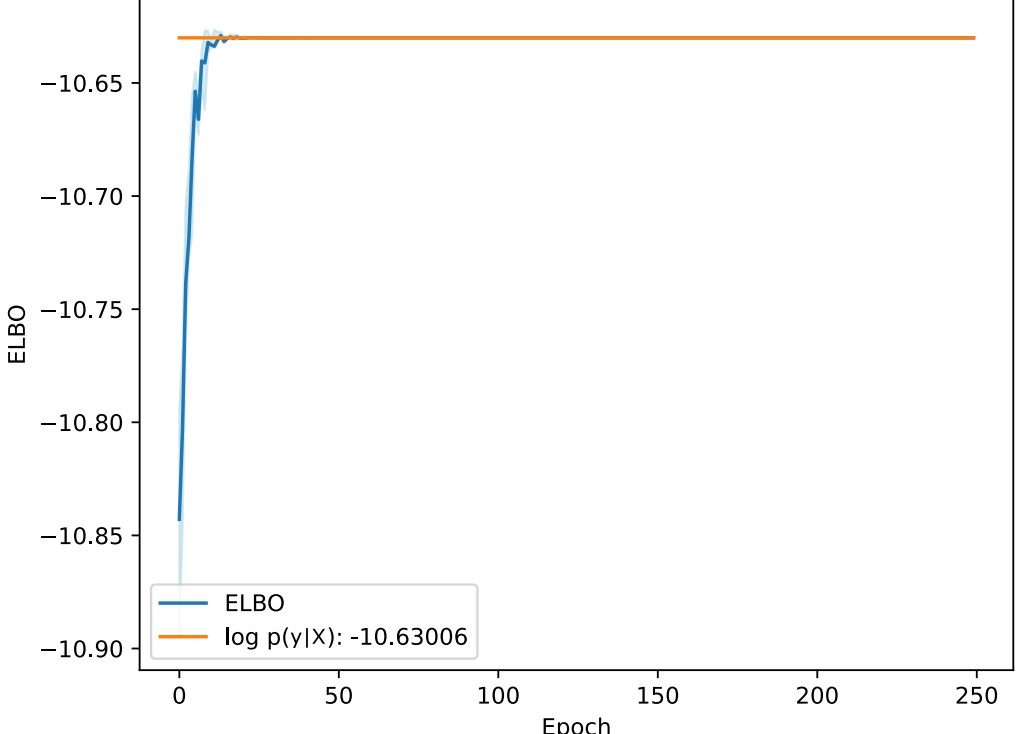

Figure 4: The median and IQR values of the ELBO (Eq. 6) over 10 runs for the simple no constant experiment where data is generated from $y = x_0$ (blue). The log of the evidence is shown in orange.

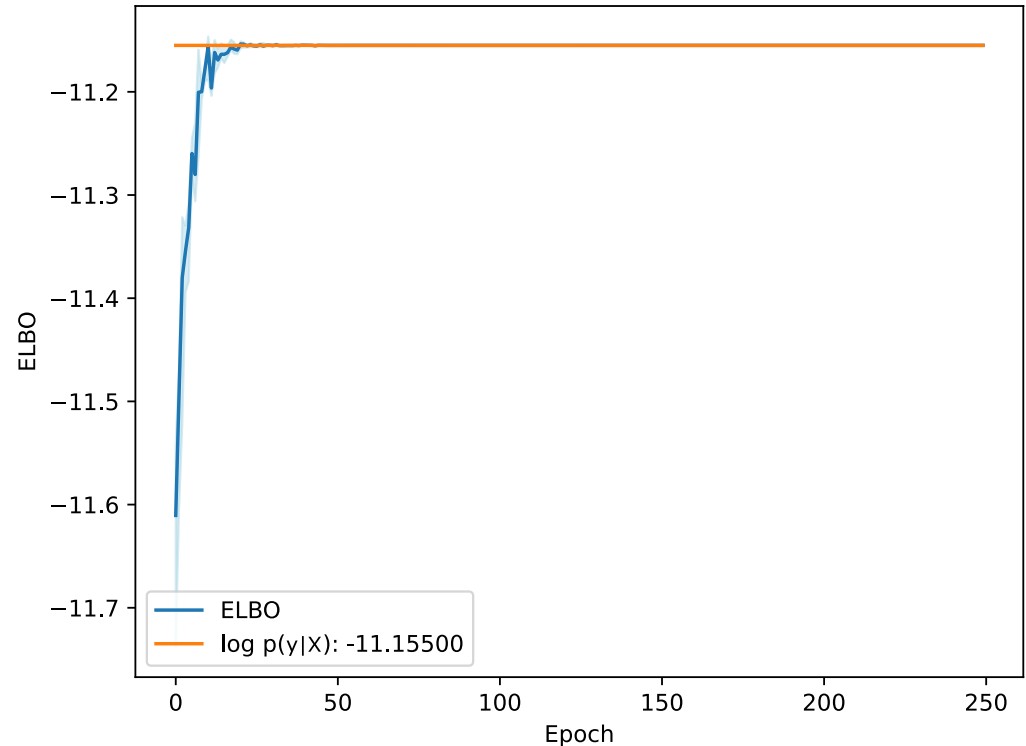

Figure 5: The median and IQR values of the ELBO (Eq. 6) over 10 runs for the simple no constant experiment where data is generated from $y = 0.5$ (blue). The log of the evidence is shown in orange.

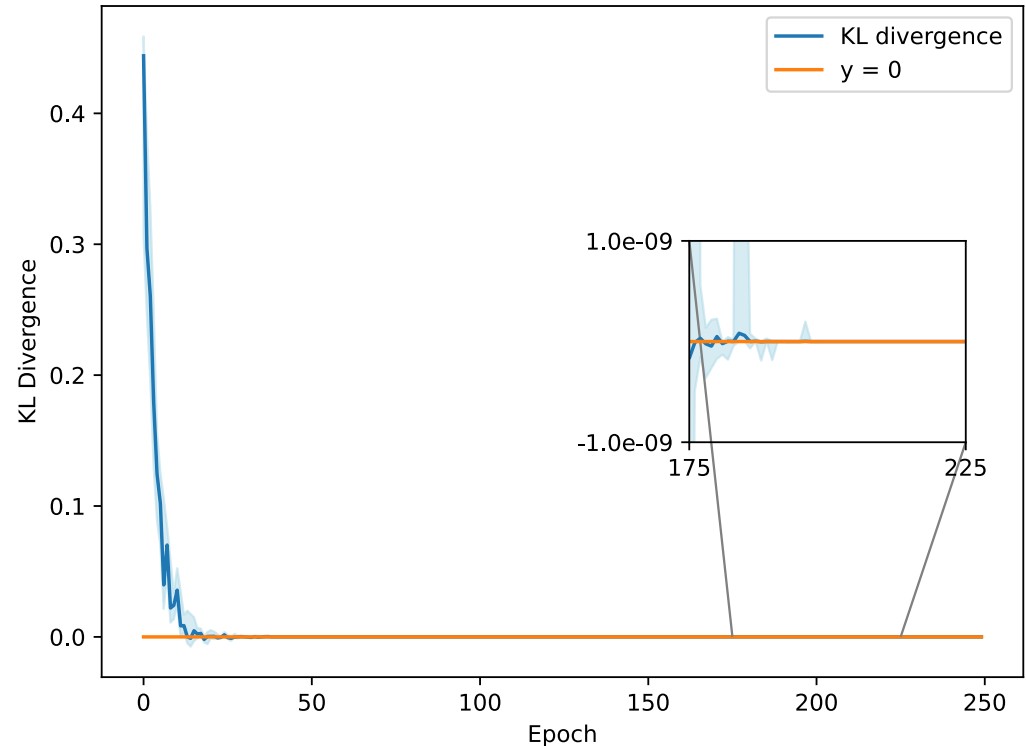

Figure 6: The median and the IQR values of the KL divergence over 10 runs for the simple no constant token experiment where data is generated from $y = x_0^2$. Magnified portion of the plot shows precision continuing to be improved near the end of the optimisation procedure. $y = 0$ is shown as a reference.

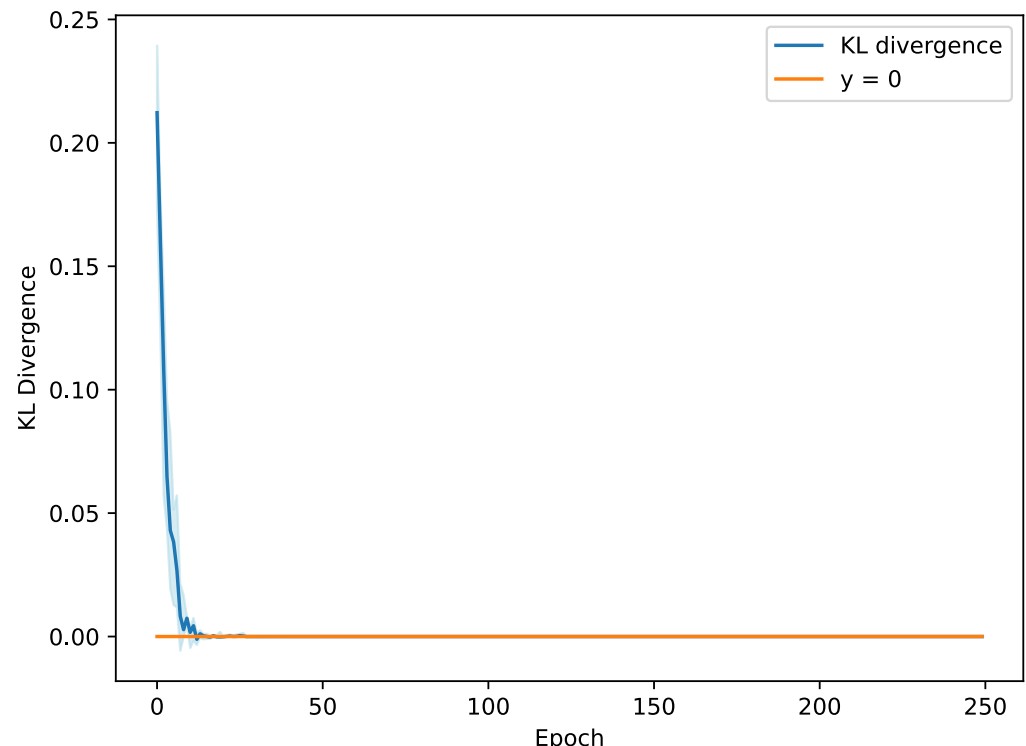

Figure 7: The median and the IQR values of the KL divergence over 10 runs for the simple no constant experiment where data is generated from $y = x_0$ (blue). The line $y = 0$ is shown in orange for reference.

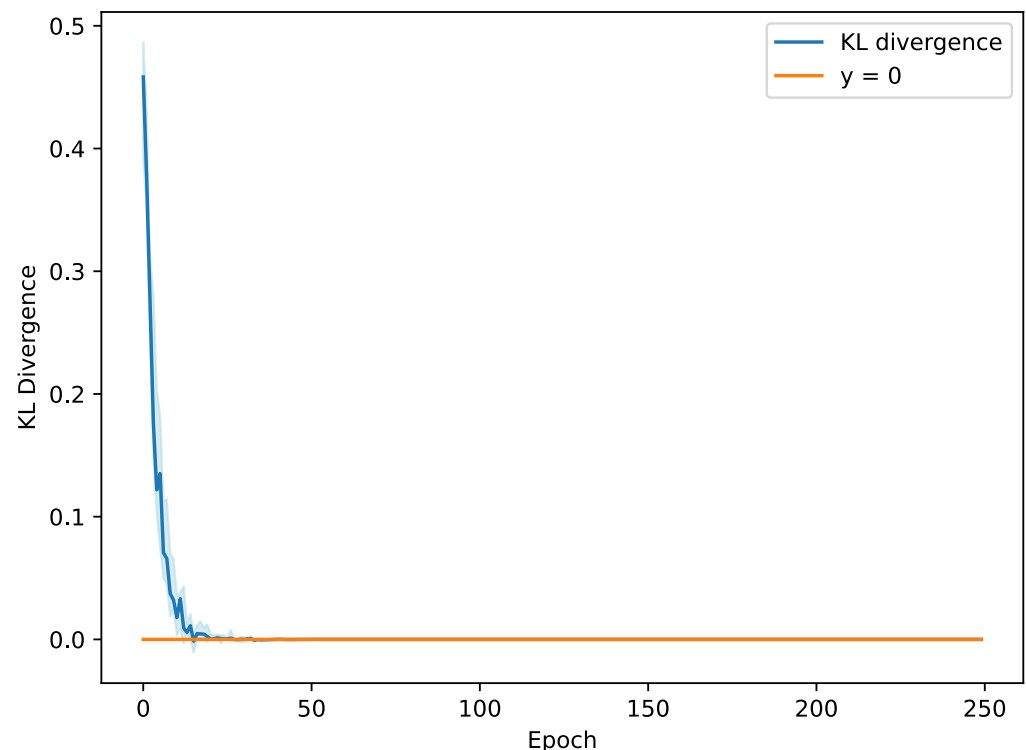

Figure 8: The median and the IQR values of the KL divergence over 10 runs for the simple no constant experiment where data is generated from $y = 0.5$ (blue). The line $y = 0$ is shown in orange for reference.

## A.5 CONSTANT TOKEN EXPERIMENTS

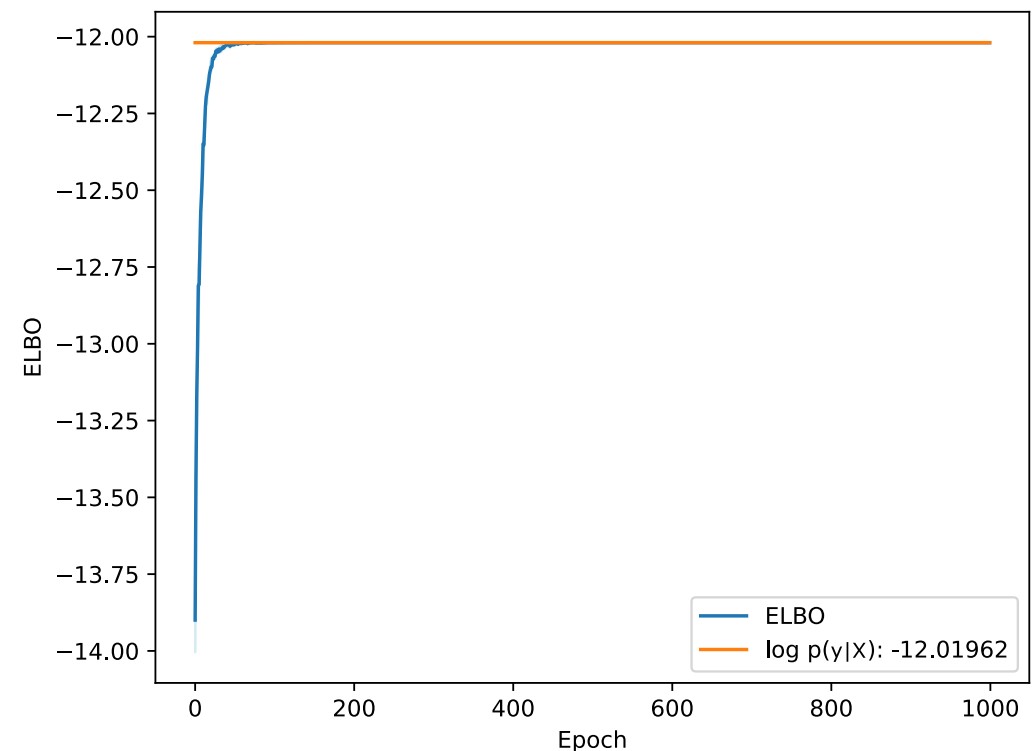

Figure 9: The median and IQR values of the ELBO (Eq. 6) over 10 runs for the simple constant token experiment where data was generated from $y = x_0^2$ (blue). The log of the evidence is shown in orange.

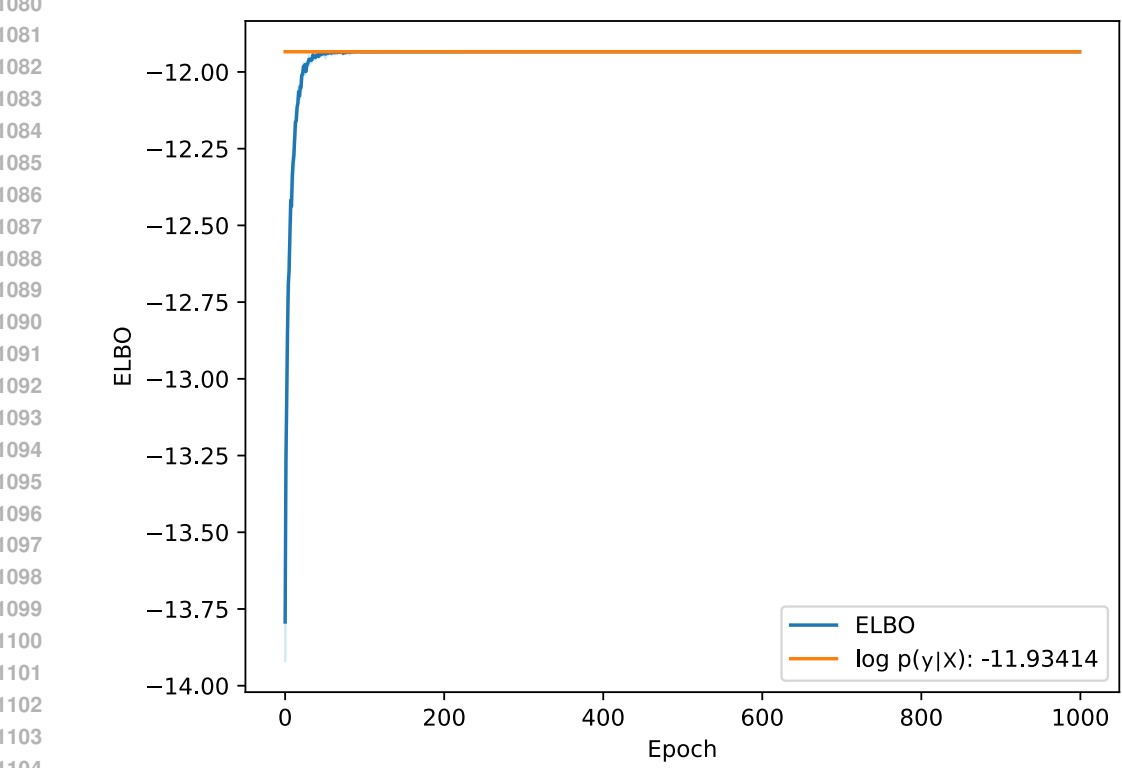

Figure 10: The median and IQR values of the ELBO (Eq. 6) over 10 runs for the simple constant token experiment where data was generated from $y = x_0$ (blue). The log of the evidence is shown in orange.

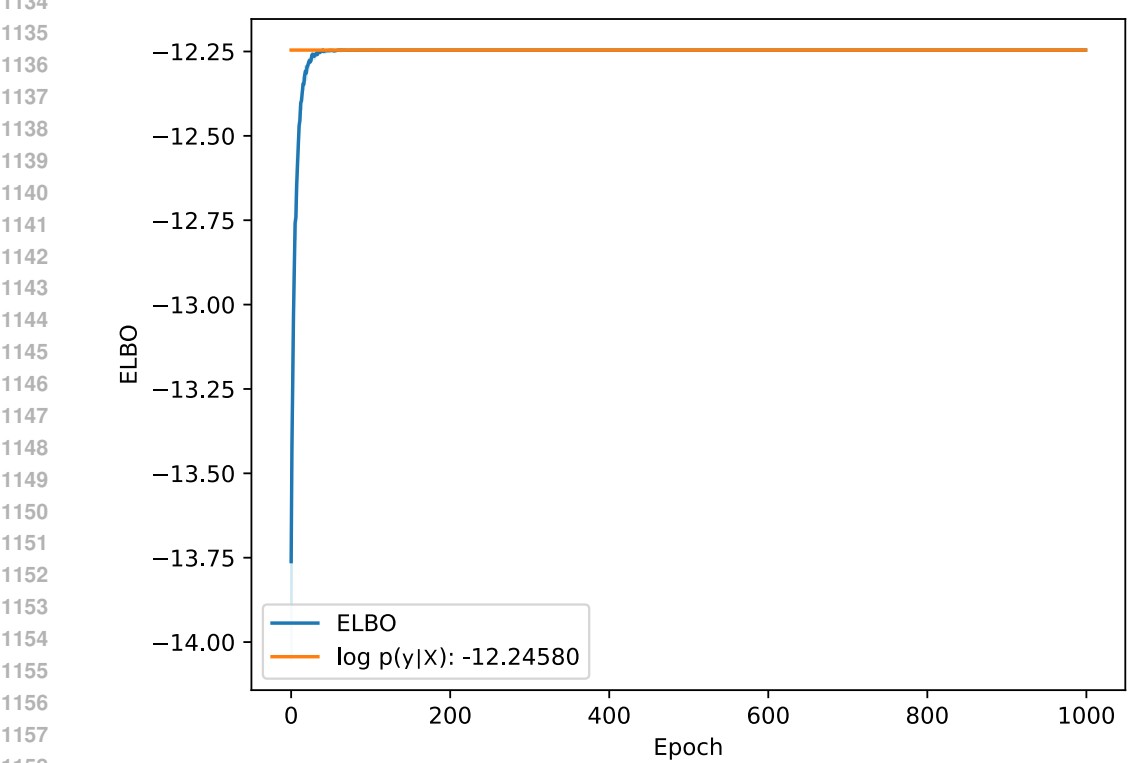

Figure 11: The median and IQR values of the ELBO (Eq. 6) over 10 runs for the simple constant token experiment where data was generated from $y = 0.5$ (blue). The log of the evidence is shown in orange.

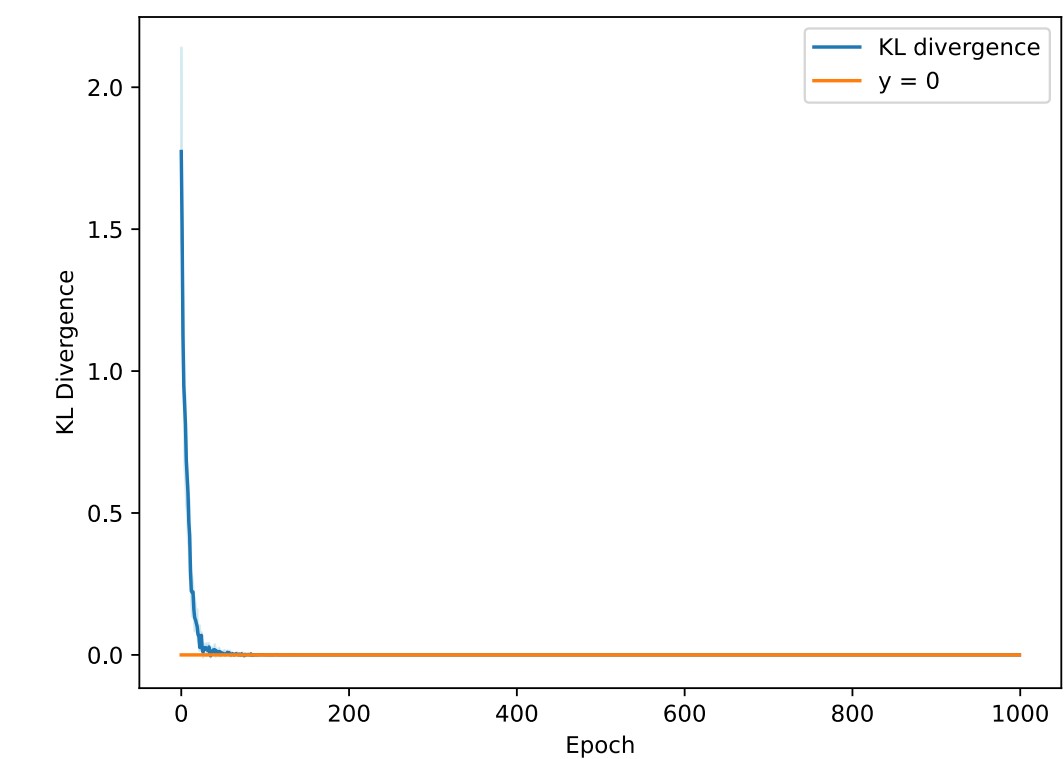

Figure 12: The median and the IQR values of the KL divergence over 10 runs for the simple constant token experiment where data was generated from $y = x_0^2$. The line $y = 0$ is shown in orange for reference.

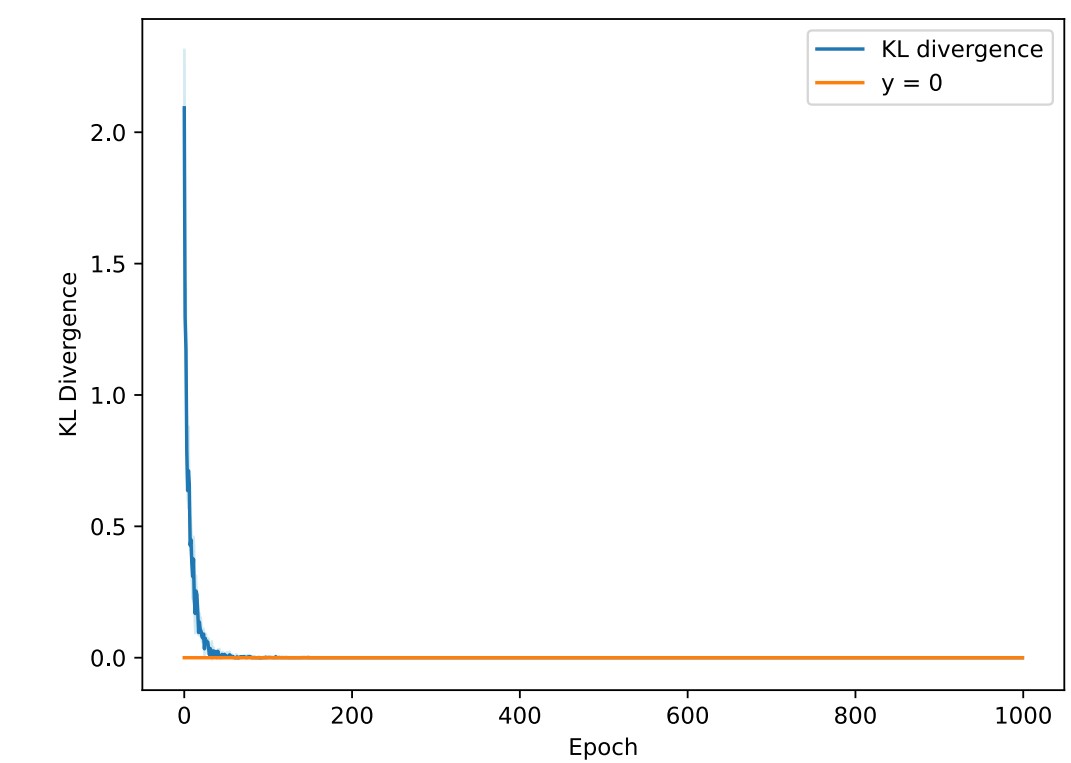

Figure 13: The median and the IQR values of the KL divergence over 10 runs for the simple constant token experiment where data was generated from $y = x_0$. The line $y = 0$ is shown in orange for reference.

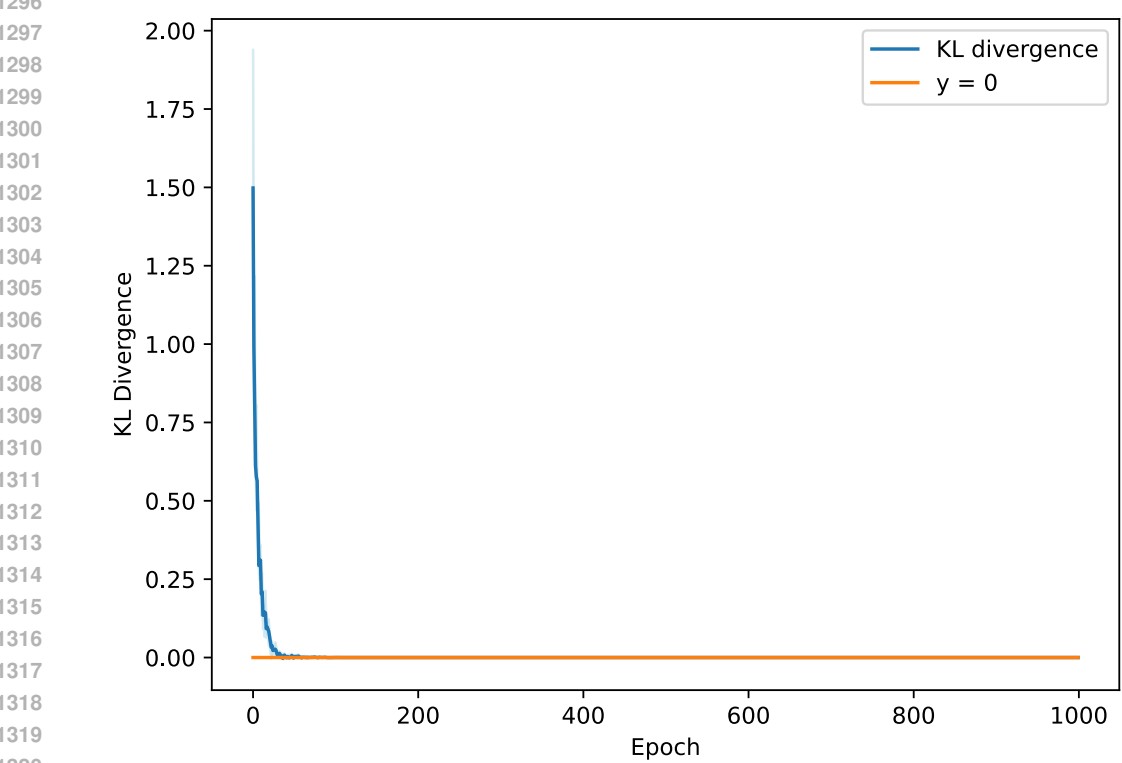

Figure 14: The median and the IQR values of the KL divergence over 10 runs for the simple constant token experiment where data was generated from $y = 0.5$. The line $y = 0$ is shown in orange for reference.

## A.6 EXPERIMENT HYPERPARAMETERS

Table 7: Hyperparameters for the three simple *no* constant token experiments.

| Hyperparameter | Value |
| --- | --- |
| Epochs | 250 |
| Samples per epoch | 100 |
| Runs | 10 |
| Max number of tokens per expression | 3 |
| RNN type | Gated Recurrent Unit |
| Number of hidden layers | 1 |
| Hidden layer size | 32 |
| Optimiser | RMSprop |
| RMSprop learning rate | 1e-2 |
| RMSprop $\alpha$ | 0.9 |
| RMSprop $\epsilon$ | 1e-6 |
| Learning rate annealer (LRA) | ReduceLROnPlateau |
| LRA metric | -ELBO |
| LRA mode | min |
| LRA factor | 0.5 |
| LRA patience | 15 |
| LRA min_lr | 1e-6 |
| Baseline | Exponential weighted moving average |
| Baseline (EWMA) $\alpha$ | 0.25 |
| Likelihood standard deviation | 1.0 |

Table 8: Hyperparameters for the scaling *no* constant token experiments.

| Hyperparameter | Value |
| --- | --- |
| Epochs | 2000 |
| Samples per epoch | 1000 |
| Runs | 1 |
| RNN type | Gated Recurrent Unit |
| Number of hidden layers | 1 |
| Hidden layer size | 64 |
| Optimiser | RMSprop |
| RMSprop learning rate | 5e-3 |
| RMSprop $\alpha$ | 0.9 |
| RMSprop $\epsilon$ | 1e-6 |
| Learning rate annealer (LRA) | ReduceLROnPlateau |
| LRA metric | -ELBO |
| LRA mode | min |
| LRA factor | 0.5 |
| LRA patience | 50 |
| LRA min_lr | 5e-6 |
| Baseline | Exponential weighted moving average |
| Baseline (EWMA) $\alpha$ | 0.25 |
| Likelihood standard deviation | 1.0 |

Table 9: Hyperparameters for the three constant token experiments.

| Hyperparameter | Value |
| --- | --- |
| Epochs | 1000 |
| Samples per epoch | 500 |
| Runs | 10 |
| Max number of tokens per expression | 3 |
| RNN type | Gated Recurrent Unit |
| Number of hidden layers | 1 |
| Hidden layer size | 64 |
| Optimiser | RMSprop |
| RMSprop learning rate | 5e-3 |
| RMSprop $\alpha$ | 0.9 |
| RMSprop $\epsilon$ | 1e-6 |
| Learning rate annealer (LRA) | ReduceLROnPlateau |
| LRA metric | -ELBO |
| LRA mode | min |
| LRA factor | 0.5 |
| LRA patience | 25 |
| LRA min_lr | 1e-6 |
| Baseline | Mean |
| Likelihood standard deviation | 1.0 |
| $c$ prior mean | 0.0 |
| $c$ prior standard deviation | 10.0 |

