# OpenReview forum: "Deep Variational Inference Symbolic Regression"
_ICLR.cc/2026/Conference — Submitted to ICLR 2026_

### Official Review · Reviewer_fEfd · 2025-10-26

**Soundness:** 3
**Presentation:** 3
**Contribution:** 2
**Rating:** 2
**Confidence:** 3

**Summary:**

The paper introduces a new algorithm, termed Deep Variational Inference Symbolic Regression (DVISR), for conducting Bayesian inference for symbolic regression. The algorithm leverages recursive neural networks (RNNs) to learn a distribution over the expression tree of mathematical expressions. DVISR is able to learn a distribution over both the constants, as well as, the expression structure. Empirical results are demonstrated on a small set of synthetic datasets coming from known mathematical expressions.

**Strengths:**

There is a growing interest in symbolic expression and developing methods that ensure that we can capture the uncertainty of the expression structure in addition to the parameter certainty seems worthwhile. The paper in general is well written and the method well motivated.

**Weaknesses:**

There are two primary weaknesses which cause me to recommend rejection at this stage: 1) weak empirical validation of DVISR, and 2) insufficient discussion of related work/consideration of relevant work.

**Weak Empirical Evaluation**

The empirical evaluation of this algorithm is limited solely to synthetic datasets with a maximum of 11 data points and data generated from very simple expressions.  The Section 3.1.1 experiment only considers 4 candidate expressions for which we can analytically compute the posterior, therefore this problem setting doesn't require a very complex algorithm in the first place. Section 3.1.2 considers more candidate expressions but as far as I can tell there is no evidence that the algorithm actually converges on the true expression. The final experiment in Section 3.2 is the only experiment that considers doing inference over both the form of the expression and the constants within the expression, but this section only considers 7 candidate expression.

*Overall, there is no experiment which tests the algorithm on real-world data, no experiment that could not be feasibly solved with a naive brute-force approach and, in general, no comparison to existing baselines.* While the paper explicitly mentions the experimental validation as a weakness of the paper, the current evaluation provides me with very little useful evidence of whether I would actually want to use this algorithm on a real-world dataset.

**Insufficient discussion of existing work**

The research field of probabilistic programming has a wide history of deriving general purpose Bayesian inference algorithms for arbitrary probabilistic programs, sometimes referred to as universal probabilistic programming [1,2]. The symbolic regression problem can be framed as inference in a universal probabilistic program and there is a wealth of papers with empirical results for the task of symbolic regression, also sometimes referred to as function induction or program synthesis [2,3,4,5,6] (this is not necessarily an exhaustive list). In general, the algorithms have been shown to scale to be able to do inference in 100s or even 1000s of expression tree while also inferring the constants and applying the algorithms to real-world data. Many of these algorithms are based on ideas from variational inference [2,3,8,9,10,11], so they could be used as relevant baselines to compare DVISR against. Overall, the probabilistic programming algorithms aim to solve a more general inference problem formulation than the symbolic approach, so I definitely see scope for deriving inference algorithms specifically targeted towards symbolic regression. However, the paper should at least discuss how the DVISR approach compares against these more general algorithms.


[1] Van De Meent, Jan-Willem, et al. "An introduction to probabilistic programming." arXiv preprint arXiv:1809.10756 (2018).

[2] Le, Tuan Anh, Atilim Gunes Baydin, and Frank Wood. "Inference compilation and universal probabilistic programming." Artificial Intelligence and Statistics. PMLR, 2017.

[3] Reichelt, Tim, Luke Ong, and Thomas Rainforth. "Rethinking variational inference for probabilistic programs with stochastic support." Advances in Neural Information Processing Systems 35 (2022): 15160-15175.

[4] Reichelt, Tim, Luke Ong, and Tom Rainforth. "Beyond Bayesian Model Averaging over Paths in Probabilistic Programs with Stochastic Support." International Conference on Artificial Intelligence and Statistics. PMLR, 2024.

[5] Zhou, Yuan, et al. "Divide, conquer, and combine: a new inference strategy for probabilistic programs with stochastic support." International Conference on Machine Learning. PMLR, 2020.

[6] Saad, Feras A., et al. "Bayesian synthesis of probabilistic programs for automatic data modeling." Proceedings of the ACM on Programming Languages 3.POPL (2019): 1-32.

[7] Saad, Feras, et al. "Sequential Monte Carlo learning for time series structure discovery." International Conference on Machine Learning. PMLR, 2023.

[8] Becker, McCoy R., et al. "Probabilistic programming with programmable variational inference." Proceedings of the ACM on Programming Languages 8.PLDI (2024): 2123-2147.

[9] Ritchie, Daniel, Paul Horsfall, and Noah D. Goodman. "Deep amortized inference for probabilistic programs." arXiv preprint arXiv:1610.05735 (2016).

[10] Harvey, William, et al. "Attention for inference compilation." arXiv preprint arXiv:1910.11961 (2019).

[11] Baydin, Atilim Güneş, et al. "Etalumis: Bringing probabilistic programming to scientific simulators at scale." Proceedings of the international conference for high performance computing, networking, storage and analysis. 2019.

**Questions:**

None.

---

> ### Author Response · Authors · 2025-11-19
>
> We first want to thank reviewer fEfd for taking time to read over our paper. Despite the fact they did not ask any questions, we wanted to comment on the suggested weaknesses.
>
> **Weak Empirical Evaluation**
>
> We certainly understand the reviewers comments here and agree that a comparison to baselines on standard benchmarks is the next step to making this work more convincing.
> Our aim with this paper was to introduce the algorithm to the community and to initially establish it was valid by showing it does in fact converge to the true posterior on very simple settings.
> Although the experiments are simple, we do think that the algorithm itself and the initial results are still interesting for members of the community.
>
> The aim of the experiments in Section 3.1.1 and Section 3.2 were not necessarily to convince the reader that our algorithm can produce complex models but simply to demonstrate that the proposed algorithm converges accurately and precisely to the true posterior.
>
>
> **Insufficient discussion of existing work**
>
> Prior to this review we were less familiar with the field of probabilistic programming, so we thank the reviewer for the references and agree that this field is certainly tangential to our work.
> In follow up work, we will make sure to embed our work within the field of probabilistic programming and be sure to experimentally compare some of the most relevant algorithms to DVISR.

---

> > ### Comment · Reviewer_fEfd · 2025-11-24
> > **Reviewer Response**
> >
> > Thank you to the authors for their response.
> >
> > I appreciate that the experiments are meant to be a first step in showing that the algorithm converges to the correct posterior. However, for a full conference submission I would expect some validation on more realistic datasets.
> >
> > Given that the other reviewers have raised similar concerns, I will maintain my current score.

---

### Official Review · Reviewer_8taT · 2025-10-28

**Soundness:** 3
**Presentation:** 3
**Contribution:** 3
**Rating:** 2
**Confidence:** 4

**Summary:**

This paper presents Deep Variational Inference Symbolic Regression (DVISR), a Bayesian extension of Deep Symbolic Regression (DSR) that aims to approximate the posterior distribution over symbolic expressions. The idea of combining variational inference with neural-guided symbolic regression is novel and promising. However, the experimental evaluation is currently limited to simple, synthetic cases, which raises significant concerns about the method's scalability, practical utility, and advantages over existing approaches. The following points detail these concerns.

**Strengths:**

The core idea of formulating symbolic regression as a variational inference problem is a significant contribution. By redefining the training objective to maximize the Evidence Lower Bound (ELBO), the authors shift the goal from finding a single best-fit expression (as in DSR) to approximating the full posterior distribution over expressions.

**Weaknesses:**

**1. Lack of Evaluation on Standard Benchmarks and Real-World Data​**

The experimental validation is conducted exclusively on minimalistic synthetic datasets (e.g., y=x0**2, y=x0) with very small expression sizes (up to 10 tokens). The absence of tests on established symbolic regression benchmarks (e.g., SRBench, AI Feynman datasets) or any real-world datasets makes it impossible to assess the method's performance in practical, challenging scenarios. The claim of "demonstrating scaling properties" is not fully convincing without comparison to standard baselines on complex problems.

**​2. Insufficient Demonstration of Predictive Accuracy​**

The experiments successfully demonstrate that the learned variational distribution q_ϕ(f)converges to the true posterior p(f∣X,y)in controlled settings. However, symbolic regression is ultimately about finding accurate models for data. The paper fails to show that a higher posterior probability correlates with a higher fit accuracy (e.g., lower NMSE on a test set). Symbolic regression is a combinatorial optimization problem where local optimality (high probability under the policy) does not guarantee global optimality (the best-fitting expression). The evaluation should connect posterior probability to actual predictive performance.

​**3. Lack of Comparative and Ablative Analysis​**

As an extension of DSR, it is crucial to benchmark DVISR against DSR and other contemporary symbolic regression methods (e.g., GP, transformer-based approaches). The absence of such comparisons leaves the reader questioning the practical benefit of the proposed Bayesian framework.

Furthermore, the paper introduces two key modifications: a new reward function based on the ELBO and an enhanced network architecture (adding the previous token and constant values as input). An ablation study is necessary to disentangle the individual contribution of each modification to the overall performance. Without it, it is unclear which component drives any potential improvement.

**​4. Questionable Utility of Constant Prediction​**

DVISR introduces the prediction of constants within expressions, which dramatically expands the search space to be infinite. While the simple experiments show convergence of the posterior over expression trees q_ϕ(z), the paper does not adequately justify why this added complexity is beneficial. Does predicting constants alongside the expression structure lead to the discovery of more accurate or meaningful expressions compared to methods that optimize constants as a separate downstream step (like DSR)? The significant increase in search difficulty must be justified by a clear and demonstrated advantage.

**Questions:**

The questions are described in the "Weakness" section.

---

> ### Author Response · Authors · 2025-11-18
>
> We first want to thank reviewer 8taT for taking time to read over our paper. Below we address each weakness in turn.
>
> **1. Lack of Evaluation on Standard Benchmarks and Real-World Data**
>
> We completely agree with the reviewers that it is very important to test the proposed algorithms on standard benchmarks and real-world datasets - this is of course our most pressing next task. Our purpose with this paper was to introduce DVISR as an algorithm to the machine learning community with the assumption that either ourselves, or other researchers, will expand the evaluations beyond the very simple examples that we have presented here. Although the experiments are slight we believed that the community would benefit from knowing about the existence of this algorithm in its own right sooner rather than later.
>
> **2. Insufficient Demonstration of Predictive Accuracy**
>
> We agree that downstream predictive importance is certainly an important metric of success for any regression algorithm and assure the reviewer that we intend to address this in future work.
>
> **3. Lack of Comparative and Ablative Analysis**
>
> It is true that we do not compare to other symbolic regression methods; however, we believe that we are adding functionality that previous symbolic regression algorithms do not have: the ability to build a distribution over models. Therefore, we belive that a comparison would be unfair with respect to finding the singular model that best fits the data in the shortest amount of time. We would argue that our algorithm may even be desirable despite worse performance compared to other symbolic regression algorithms due to the additional advantage of uncertainty quantification. Despite this, we do intend to perform a full experimental comparison to existing symbolic regression algorithms and Bayesian symbolic regression algorithms in future work.
>
> We believe that both the reward function alteration and the proposed architectural change are necessary for a fully Bayesian solution and that leaving out either component would result in a algorithm that produced a distribution that did not converge to the true posterior - that is the reason that we did not perform ablation studies for either of these two components. However, we are open to the idea that ablation studies could be performed in relation to the model inputs and we are interested to see whether each input is necessary to acheive good performance.
>
> **4. Questionable Utility of Constant Prediction**
>
> It is true that including constants in the posterior makes the search space much larger and incurs a larger computational cost. However, to not include constants in the posterior means that you are not building a probability distribution over the _full_ model space. Then, all models that contain constants that are not the best fit constants would be assigned a probability of zero, which would be an inaccurate representation of how likely those models represented the observed data.

---

### Official Review · Reviewer_EHVz · 2025-10-31

**Soundness:** 2
**Presentation:** 3
**Contribution:** 2
**Rating:** 4
**Confidence:** 2

**Summary:**

This paper introduces Deep Variational Inference Symbolic Regression (DVISR), a novel method that extends Deep Symbolic Regression (DSR) into a fully Bayesian framework. The key innovation is the reformulation of the training objective to maximize the Evidence Lower Bound (ELBO), enabling the model to approximate the full posterior distribution over symbolic expressions rather than finding a single point estimate. The authors provide convincing evidence on small-scale, controlled problems that DVISR can successfully recover the true posterior distribution for both expression structure and constants.

**Strengths:**

The core strength of this work is its novel and well-motivated conceptual framework, which effectively combines the interpretability of symbolic regression with the uncertainty quantification of Bayesian inference, as a direct improvement of the DSR method. The theoretical grounding adds significant depth and credibility to the methodological claims.

**Weaknesses:**

The primary weakness is the limited scale and scope of the experimental validation, which only demonstrates efficacy on very small synthetic problems with minimal expression complexity and no noise. The paper does not show that the method scales to problems of practical interest or performs competitively against existing symbolic regression algorithms on standard benchmarks. Furthermore, the comparison to related work, especially other Bayesian symbolic regression approaches, is relatively superficial.

**Questions:**

1. What is the core advantage of using Bayesian probability modeling? Can you summarize its key strengths?
2. Was any ablation study conducted to evaluate the contribution of the specific architectural change, such as providing the previous token as input?

---

> ### Author Response · Authors · 2025-11-18
>
> We first want to thank reviewer EHVz for taking time to read over our paper.
>
> We recognise the concerns of the reviewer in relation to the scope of the experimental validation and agree that it is of paramount importance to next evaluate DVISR on more complex and real-world domains; however, we leave these experiments for future work.
> In the future, we also fully intend to compare our proposed algorithm against other symbolic regression algorithms.
>
> _1. What is the core advantage of using Bayesian probability modelling? Can you summarize its key strengths?_
>
> Bayesian modelling is useful for capturing both the epistemic and aleatoric uncertainties within a system. Most previous work on symbolic regression aims to locate a _single_ model that best explains the data. Whilst the single model that best fits the data is obviously valuable, it is often that case that the data has been produced by an underlying model that is _not_ the best fit to the data, but either data noise or a small number of data points gives another impression. As a result, we believe - as do many Bayesianists - that it is important to model a probability distribution over models which accounts for these issues. Another advantage of a Bayesian approach is that it naturally incorporates prior knowledge, which can be helpful for injecting domain knowledge or when operating within a small data regime.
>
>
> _2. Was any ablation study conducted to evaluate the contribution of the specific architectural change, such as providing the previous token as input?_
>
> We did not perform any architectural ablations, which is certainly important to discern the most important components and to potentially simplify the structure - we will leave this to future work. With regards to network inputs, we wanted to provide the network with all the information it needed to make adequate decisions and for Equation 1 to still hold (that the likelihood of the current token is conditioned on _all_ previous tokens and constants). However, there is certainly scope to experiment with what inputs to include. In fact, including the parent and sibling inputs is not strictly necessary for Equation 1 to hold, as long as the previous input is included; however, Petersen et al. 2021 did report improved performance when including parent and sibling input tokens.

---

> > ### Comment · Reviewer_EHVz · 2025-11-25
> > **Reply**
> >
> > Thank you very much for your reply. I think these experiments are important to the paper, and this paper would be incomplete without them. So I keep my score

---

### Official Review · Reviewer_Y6CD · 2025-11-01

**Soundness:** 3
**Presentation:** 3
**Contribution:** 2
**Rating:** 2
**Confidence:** 5

**Summary:**

The paper extends DSR by replacing its heuristic reward with the ELBO, training the policy with REINFORCE so that maximizing reward corresponds to variational inference over expressions. It also has the network output distributions for constants, yielding posteriors over both trees and parameters. Experiments are toy-scale: tiny libraries and depths (with and without constants), where the authors can compute the evidence exactly (or via controlled quadrature) and show ELBO→log-evidence and KL→0; a scaling study shows degradation as maximum expression length grows.

**Strengths:**

Clear probabilistic objective. Casting the policy objective exactly as the ELBO (not risk-seeking) makes the optimization/probabilistic story coherent and checkable.

Posteriors over constants and structures. The architecture outputs distributions for constants, so the method produces joint posteriors instead of doing separate constant fitting downstream.

Positioning vs. VI literature. The paper distinguishes DVISR from earlier VI-style SR that were limited (polynomials only, partial parameter sets).

**Weaknesses:**

Novelty vs. prior neural-policy SR is limited. Using an RL-trained neural policy to sample expression tokens is well-trod (DSR/Neural-guided GP/PhySO/FEX). The paper’s main delta is the VI objective and constant distributions. Please delineate algorithmic differences and where VI yields tangible gains beyond conceptual neatness (e.g., better calibrated uncertainty, improved exploration, or sample efficiency), ideally with head-to-head ablations against DSR/PhySO/FEX/PySR.

No runtime/efficiency comparisons. Claims of amortization aren’t backed by wall-clock or budgeted comparisons. Add training time, sample counts, and hardware vs. DSR, PhySO, PySR, FEX on standard suites (Feynman, SRBench), and report accuracy/recovery vs. time curves.

Scaling limits left unresolved. The scaling study shows KL grows with expression size; the paper itself lists decreased performance as a key limitation. Provide architectural ablations (e.g., transformers or PPO/GRPO objectives the authors mention) and demonstrate whether they actually improve scaling on the same search spaces.

Evaluation scope is narrow. Current tests are tiny and noiseless (except simple cases), with evidence computable only because the model space is tiny or integrals are defined carefully. Add real datasets (noisy data, physics with units) and report posterior calibration metrics (coverage, NLL, PIT histograms), not just convergence to an analytic posterior in toy settings.

Use of priors not compared to alternatives. If structural/physical priors are used (even lightly), compare to grammar/units prior baselines and show contribution via +prior/−prior ablations.

Presentation gap: ELBO pieces vs. practical likelihood. The likelihood assumes fixed σ and i.i.d. Gaussian noise; real-world SR often needs heteroscedasticity/robust losses. Consider likelihood sensitivity ablations: different σ, Student-t, or learned noise heads, and show how the posterior shifts.

**Questions:**

Where is VI strictly better than risk-seeking/heuristic rewards? Provide a controlled comparison showing improved uncertainty calibration, exploration diversity, or sample efficiency when optimizing ELBO vs. a standard DSR-style reward.

Runtime and budgets. What are the wall-clock times and numbers of sampled expressions to reach a given recovery/accuracy on Feynman or SRBench? Any evidence of amortization paying off vs. per-candidate constant fitting?

Posterior quality. Do you measure calibration (e.g., coverage at nominal levels) or report NLL/CRPS on held-out data? If not, please add.

Scaling mitigation. Of the proposed fixes (transformers, PPO/GRPO), which have you prototyped, and what empirical gains do they bring on the same scaling benchmark?

Constants posterior expressivity. You currently use Gaussians; do mixtures/flows help capture multimodality without exploding variance-reduction needs? Any results?

Evidence computation in “with constants.” The appendix adopts a special handling of integrals over constants not in a given tree. Please clarify the definition and numerics; can you validate it on slightly larger constant sets with importance sampling or nested quadrature?

Comparisons to Bayesian SR (MCMC/EM). Include a Bayesian baseline (e.g., BSR/MCMC) on the exact same toy settings to show that VI reaches comparable posteriors much faster—or explain why not.

---

> ### Author Response · Authors · 2025-11-19
>
> We first want to thank reviewer Y6CD for taking time to read over our paper and providing such a detailed and helpful review.
>
> Overall, we sympathise with the reviewers concerns regarding minimal experimental settings and lack of extensive comparisons and analyses.
> Our aim with this paper was to present the algorithm to the community, which we show is valid by illustrating the expected convergence in simple settings.
> Despite our experiments and analyses being limited we hoped that either ourselves, or other members of the community, would continue to scale the algorithm and provide more robust analysis in future work; and, even in its current form, we believe this work is interesting to others in the field.
>
> Below we address each of the questions in turn.
>
> **Where is VI strictly better than risk-seeking/heuristic rewards? Provide a controlled comparison showing improved uncertainty calibration, exploration diversity, or sample efficiency when optimizing ELBO vs. a standard DSR-style reward.**
>
> If we were to use a risk-seeking reward rather than the ELBO we would lose the property of convergence to the true posterior.
> Whilst we agree that a direct comparison of the ELBO to other rewards would be illuminating, we think that by providing the full posterior we are offering an _additional_ element not available when using other rewards.
> Hence, a direct comparison of a metric such as sample efficiency may make our proposed algorithm appear suboptimal; however, we would argue our algorithm provides the additional advantage of uncertainty quantification.
>
> **Runtime and budgets. What are the wall-clock times and numbers of sampled expressions to reach a given recovery/accuracy on Feynman or SRBench? Any evidence of amortization paying off vs. per-candidate constant fitting?**
>
> We agree that these metrics measured on standard SR benchmarks, such as Feynman and SRBench, are certainly important and we wish to carry out these experiments soon.
>
> **Posterior quality. Do you measure calibration (e.g., coverage at nominal levels) or report NLL/CRPS on held-out data? If not, please add.**
>
> We have not directly computed these measures but we will include them in future analyses.
>
> **Scaling mitigation. Of the proposed fixes (transformers, PPO/GRPO), which have you prototyped, and what empirical gains do they bring on the same scaling benchmark?**
>
> We have not yet prototyped alternative policy gradient methods or architectures but we anticipate that they will bring measurable improvements to the algorithm and intend to explore them soon.
>
> **Constants posterior expressivity. You currently use Gaussians; do mixtures/flows help capture multimodality without exploding variance-reduction needs? Any results?**
>
> We did not experiment with any distributions other than Gaussians but certainly agree with the reviewer that more expressive distributions could potentially improve performance.
> Given that the exact posterior was recovered in our simple experiments, we took this as evidence that Gaussians were already expressive enough in these simple cases.
> However, we think it is likely that for more complex functions solely using Gaussians may restrict convergence and incorporating your suggestion will be necessary.
>
> **Evidence computation in “with constants.” The appendix adopts a special handling of integrals over constants not in a given tree. Please clarify the definition and numerics; can you validate it on slightly larger constant sets with importance sampling or nested quadrature?**
>
> We do treat this specific integral in a non-intuitive way but we believe that our proof in Appendix A.2 does in fact demonstrate that this handling results in the correct evidence calculation in _all_ cases.
> We think that illustrating its correctness on larger constant sets would only serve to demonstrate the effectiveness of the numerical integrator component (on those constants _within_ the tree) rather than illustrating the correctness of the integral for constants _not_ in the tree.
>
> **Comparisons to Bayesian SR (MCMC/EM). Include a Bayesian baseline (e.g., BSR/MCMC) on the exact same toy settings to show that VI reaches comparable posteriors much faster—or explain why not.**
>
> We intend to perform these comparison in future work and agree it is certainly important to back up our claim that one advantage of using VI is faster convergence compared to MCMC.

---

> > ### Comment · Reviewer_Y6CD · 2025-11-28
> >
> > I would like to follow up specifically on the statement in the rebuttal that “If we were to use a risk-seeking reward rather than the ELBO we would lose the property of convergence to the true posterior.” As written, this sounds like (i) a general disadvantage of risk-seeking rewards in optimization and (ii) a claim that does not align with my understanding of existing neural-policy SR methods.
> >
> > In the symbolic regression literature, many approaches use neural networks to parametrize a probability distribution over operators/tokens, sample expressions from this distribution, and then train via risk-seeking rewards (e.g., based on fit error or other heuristic scores). In these settings, one does not typically observe a fundamental “non-convergence” issue attributable solely to the risk-seeking nature of the reward.
> >
> > If your claim is that in your specific formulation a risk-seeking objective cannot be written as a valid ELBO and hence does not guarantee convergence to the Bayesian posterior (in the sense of variational inference), then I think this needs to be stated much more precisely. In particular, it would be helpful if you could clarify:
> >
> > What precise notion of “convergence to the true posterior” you are using (e.g., convergence in KL, almost sure convergence of samples, or something else);
> >
> > In what sense a risk-seeking reward breaks this property mathematically in your setup (e.g., which term in the ELBO is no longer optimized, or how the risk-sensitive objective deviates from the variational objective);
> >
> > Whether your claim is meant to be a general statement about risk-seeking rewards in SR, or a statement specific to your ELBO-based formulation.
> >
> > Right now, the rebuttal reads as if risk-seeking rewards are inherently non-convergent in a broad sense, which I do not find convincing without a more formal justification. A sharper, mathematically grounded comparison between the ELBO objective and a standard risk-seeking SR reward would strengthen the paper considerably.
> >
> > Since there are many important experiments to be done and report by the authors in the future, I'd like to keep my original score and hope the authors can add those results next time in a new submission.

---

> > > ### Author Response · Authors · 2025-12-01
> > >
> > > Firstly, thank you asking for clarification about this particular point. We understand that we may not have been clear in the main paper about specifically why we never used a risk-seeking reward.
> > >
> > > We are certainly not claiming a disadvantage of risk-seeking rewards in optimization in general, we do understand the advantages that they bring - one example being in the Deep Symbolic Regression paper that our work was based on. However, in our work, we cannot simply optimize the proposed distribution according to the top epsilon performing samples without altering the required convergence property.
> > >
> > > As we understand it, in order to successfully perform variational inference one needs to maximise the ELBO, which is an expectation over the proposed distribution. In this work, we estimate the gradient of this expectation by sampling a batch of N samples from the proposed distribution and computing an unbiased estimator (eq. 4), which approximates the true gradient of the ELBO (eq. 7). If we were, however, only to use a subset of those N samples that achieved the highest reward, then the gradient estimator would be biased and we would in fact be optimising an alternative objective rather the ELBO.
> > >
> > > In future work, we will make this much more clear and provide a more in depth justification of why we do not use a risk-seeking reward like the previous works that our work is based on.

---

### Meta-Review · Area_Chair_8aig · 2025-12-17

**Summary:**

This paper introduces Deep Variational Inference Symbolic Regression (DVISR), a Bayesian extension of Deep Symbolic Regression (DSR). The core idea is to replace DSR’s heuristic or risk-seeking reward with an objective derived from the Evidence Lower Bound (ELBO), thereby framing symbolic regression as a variational inference problem over symbolic expressions. The proposed method uses a neural policy to define a variational distribution over expression trees and, unlike prior DSR-style approaches, also outputs distributions over numerical constants within expressions, enabling joint posterior inference over structure and parameters. The authors provide a clear probabilistic formulation and show, through carefully constructed small-scale experiments, that DVISR can converge to the true posterior in settings where the evidence is analytically or numerically tractable. Additional experiments explore how performance degrades as the maximum expression size increases.

Reviewers agree that the paper presents a conceptually clean and principled formulation of symbolic regression as variational inference. The use of the ELBO as the optimization objective is theoretically well motivated and clearly distinguished from risk-seeking policy-gradient approaches used in prior work. The extension to modeling posterior distributions over both expression structure and constants is novel and technically sound.

Overall, the reviewers are aligned in viewing this work as methodologically promising but empirically underdeveloped for acceptance at ICLR. While the theoretical formulation and small-scale validations are sound, the current submission falls short of the experimental depth and practical demonstration expected at this venue. The consensus recommendation is reject, with encouragement to resubmit after substantially expanding the experimental evaluation to include medium-scale problems, realistic or noisy datasets, and comparisons to relevant baselines. The methodological contribution appears valuable, but stronger empirical evidence is needed to justify its impact and practical relevance.

**Reviewer Concerns:**

The main limitation, consistently identified by all reviewers, is the very limited experimental scope. All evaluations are restricted to small, synthetic problems with minimal expression depth, few variables, and mostly noiseless data. There are no experiments on standard symbolic regression benchmarks, real-world datasets, or moderately complex expressions where the evidence cannot be computed exactly.

As a result, the paper does not convincingly demonstrate:

1) Practical scalability of the approach,

2) Empirical advantages over existing symbolic regression methods (including DSR),

3) Benefits of the Bayesian formulation in terms of uncertainty calibration, predictive performance, or runtime efficiency.

Additionally, the lack of comparative baselines, ablation studies, and runtime analysis makes it difficult to assess the practical impact of the proposed method. While the paper acknowledges scaling limitations, no empirical mitigation strategies are explored.

The authors’ rebuttal is thoughtful and technically accurate. They clarify the relationship between ELBO optimization and risk-seeking rewards, and they appropriately acknowledge the reviewers’ concerns regarding experimental breadth and comparisons. However, most responses defer requested analyses and experiments to future work, and no new empirical evidence is provided during the rebuttal period. Consequently, the rebuttal does not materially resolve the core concerns raised by the reviewers.

**Reviewer Scores:**

The response is unlikely to have changed any of the scores.

---

### Decision · Program_Chairs · 2026-01-26

Reject